# Energy and Climate Policy—An Evaluation of Global Climate Change Expenditure 2011–2018

**Coilín ÓhAiseadha [1,\*], Gerré Quinn [2] , Ronan Connolly [3,4] , Michael Connolly [3] and Willie Soon [4]**

1   Department of Public Health, Health Service Executive, Dr Steevens' Hospital, D08 W2A8 Dublin 8, Ireland
2   Centre for Molecular Biosciences, Ulster University, Coleraine BT521SA, Northern Ireland, UK; g.quinn@ulster.ac.uk
3   Independent Scientists, Dublin 8, Ireland; ronan@ceres-science.com (R.C.); michael@ceres-science.com (M.C.)
4   Center for Environmental Research and Earth Sciences (CERES), Salem, MA 01970, USA; willie@ceres-science.com
*   Correspondence: coilin.ohaiseadha@hse.ie

**Abstract:** Concern for climate change is one of the drivers of new, transitional energy policies oriented towards economic growth and energy security, along with reduced greenhouse gas (GHG) emissions and preservation of biodiversity. Since 2010, the Climate Policy Initiative (CPI) has been publishing annual Global Landscape of Climate Finance reports. According to these reports, US$3660 billion has been spent on global climate change projects over the period 2011–2018. Fifty-five percent of this expenditure has gone to wind and solar energy. According to world energy reports, the contribution of wind and solar to world energy consumption has increased from 0.5% to 3% over this period. Meanwhile, coal, oil, and gas continue to supply 85% of the world's energy consumption, with hydroelectricity and nuclear providing most of the remainder. With this in mind, we consider the potential engineering challenges and environmental and socioeconomic impacts of the main energy sources (old and new). We find that the literature raises many concerns about the engineering feasibility as well as environmental impacts of wind and solar. However, none of the current or proposed energy sources is a "panacea". Rather, each technology has pros and cons, and policy-makers should be aware of the cons as well as the pros when making energy policy decisions. We urge policy-makers to identify which priorities are most important to them, and which priorities they are prepared to compromise on.

**Keywords:** climate mitigation; climate adaptation; renewable energy; solar energy; wind energy; biomass; biofuels; e-vehicles; energy poverty; energy justice

## 1. Introduction

In view of changes to the world climate system since the 1950s, the United Nations' Intergovernmental Panel on Climate Change (IPCC) has concluded that continued emission of greenhouse gases (GHG) will cause "further warming and long-lasting changes in all components of the climate system, increasing the likelihood of severe, pervasive and irreversible impacts for people and ecosystems" (p8, IPCC Synthesis Report (2014)) [1]. This conclusion, along with the conclusions of the UN Framework Convention on Climate Change (UNFCCC), has inspired the ongoing efforts of the UN Conference of the Parties (COP) since the 1990s to coordinate international agreements to urgently and substantially reduce greenhouse gas emissions, such as the 1996 Kyoto Protocol [2] and the 2015 Paris Agreement [3].

The efforts that have been invested in achieving agreement on these major international negotiations are a remarkable testament to international concern and support for these goals. However, greenhouse gas emissions have continued to rise [4–6]. A key underlying problem is that most of the rise in greenhouse gas emissions (chiefly carbon dioxide, $CO_2$) since the 19th century is due to the use of fossil fuel-generated energy (coal, oil, natural gas, and peat), which has driven the Industrial Revolution [7]. This cheap and abundant energy has facilitated unprecedented increases in standards of living, average lifespan, technological advances, agriculture, and world population along with economic growth [7–9]. It is clear that, historically, it was a key factor in enabling the development of the current high-income nations [7–9]. Gupta (2014) noted that this has been a major source of contention between developing and developed nations in international attempts to reduce global greenhouse gas emissions [10]. Specifically, if developing nations follow the same well-tested path that nations have historically taken to become developed, this would dramatically increase greenhouse gas emissions, and this raises a debate as to whether international treaties to reduce greenhouse gas emissions are implicitly hindering the development of developing nations [10].

On the other hand, several researchers and opinion-makers have argued that a "zero-carbon" alternative post-industrial revolution, involving a transition towards wind- and solar-generated electricity, along with the widespread electrification of transport systems and improvements in energy efficiency (possibly also including bioenergy) is not only feasible, but desirable, e.g., Gore (2006, 2017) [11,12], Jacobson et al. (2011, 2015, 2017, 2018) [13–16], Klein (2015) [17], and Goodall (2016) [18]. Although these claims have been disputed in the scientific literature [19–24], they are eagerly promoted by environmental advocacy groups such as Greenpeace [25,26] and protest movements such as "Extinction Rebellion" [27] and "Fridays For Future" [28], achieving strong currency in both mainstream and social media. This has prompted many political groups and governments to reshape their policy platforms accordingly [29,30], e.g., in terms of a "Green New Deal" [31–33].

Given the popularity of this framing, it is unsurprising that many people assume that opposition to these policies arise from ignorance, a lack of concern for the environment, and/or the lobbying of vested interests calling for business as usual [34–37]. However, much of the opposition is voiced by environmentalists and researchers who are concerned about environmental and societal problems associated with these policies as well as the lack of critical discussion of the engineering and economic feasibility of these policies [8,20,26,38–45].

Many criticisms of these "zero-carbon" proposals arise from simple engineering and economic practicalities. Some have questioned whether the proposed "green technologies" are able to meet the energy demands of the current population, let alone an increasing population [20,38,41,43,44,46,47]. For example, from an evaluation of 24 studies of 100% renewable electricity, Heard et al. (2017) found that, "based on our criteria, none of the 100% renewable electricity studies we examined provided a convincing demonstration of feasibility" [21]. A major engineering problem with wind-, solar-, and also tidal-generated electricity is that these are "intermittent" (also called "non-dispatchable" or "variable") electricity generation technologies. While it has been argued that this can in principle be overcome through a combination of energy storage [48,49] and/or a major continental-scale expansion in the electricity transmission networks [50], others have noted that the scale of these projects is enormous [19,21–24,45]. Many have asked why, if reducing greenhouse gas emissions is to be genuinely considered as the top priority, solutions involving increases in nuclear energy and/or transitioning from coal/oil to natural gas are continually dismissed or sidelined [20,21,23,38,39,41–44,51,52]?

Ironically, given that these policies are framed as being environmentally desirable, many of the criticisms are with their environmental impacts. Many researchers are concerned about the negative impacts that "green energies" have on biodiversity [51,53–56]. Some have noted that the transition to these technologies would require a huge increase in the mining of limited resources [45,57,58], with Mills (2020) arguing that, "Compared with hydrocarbons, green machines entail, on average, a 10-fold increase in the quantities of materials extracted and processed to produce the same amount of energy" [45]. Some note that large-scale wind farms can cause significant *local* climate change

(as distinct from the *global* climate change from greenhouse gas emissions they are purported to be reducing) [59–66].

Pielke Jr. (2005) notes that there are two approaches to reducing the impacts of future climate change: (i) "climate mitigation" and (ii) "climate adaptation" [67]. The first approach, "climate mitigation", assumes that greenhouse gases are the primary driver of climate change and tries to "reduce future climate change" by reducing greenhouse gas emissions. The second approach, "climate adaptation", involves developing better systems and infrastructure for dealing with climate change and climate extremes. Pielke Jr. argues that by overemphasizing "climate mitigation", the UNFCCC and the COP agreements, such as the Kyoto Protocol (and more recently the Paris Agreement), have created a bias against investment in climate adaptation. He also notes that climate mitigation policies explicitly assume that climate change is primarily driven by greenhouse gas emissions, whereas climate adaptation policies often make sense regardless of the causes of climate change. With that in mind, it is worth noting that several recent studies have argued that the IPCC reports have underestimated the role of natural factors in recent climate change (and hence overestimated the role of human-caused factors) [68–71].

Furthermore, in this Special Issue of Energies, Connolly et al. (2020) have noted that, even assuming climate change is primarily due to human-caused greenhouse gas emissions, the amount of global warming expected under business-as-usual policies is heavily determined by a metric called the "climate sensitivity" [6]. The exact value of this metric is the subject of considerable ongoing scientific debate, but Connolly et al. calculated that, if the value is at the higher end of the IPCC's range of estimates, then we can expect that the Paris Agreement's stated goal of keeping human-caused global warming below 2 °C will be broken under business as usual by the mid-21st century, whereas, if the climate sensitivity is at the lower end of the IPCC's estimates, then the Paris Agreement will not be broken under business-as-usual until at least the 22nd century. In other words, they showed that the scientific community has still not satisfactorily resolved whether reducing greenhouse gas emissions is a problem for this century or the next. This has implications for establishing exactly how urgent the proposed transitions to "low-carbon" policies are. This is important because, notwithstanding concern over the climate change which the associated greenhouse gas emissions might be causing, the existing fossil fuel-driven energy policies have many benefits [8,9]. Indeed, it is worth noting that the main greenhouse gas of concern, carbon dioxide ($CO_2$), is a key component of all carbon-based life, i.e., all known life, and that increasing atmospheric carbon dioxide concentrations have contributed to a partial "greening of the Earth", i.e., increased plant growth over the last few decades [9,72].

In light of the above criticisms, the reader may wonder whether the current proposed "zero-carbon" energy transition policies based predominantly on wind- and solar-generated electricity are truly the panacea that promoters of these technologies indicate [11–18,25,27,28]. This is a key question which we aim to address in this review paper. We hope that, by the end of this review, the reader will appreciate that none of the current energy and electricity sources used by society are a "panacea". Rather, each technology has its pros and cons, and policy-makers should be aware of the cons as well as the pros when making energy policy decisions. We urge policy-makers to identify which priorities are most important to them, and which priorities they are prepared to compromise on. Sovacool and Saunders (2014) [73] provide a useful framework for this by comparing and contrasting five different energy security policy packages. They found that all five packages have advantages and disadvantages, and that "energy security is not an absolute state, and that achieving it only 'works' by prioritizing some dimensions, or policy goals and packages, more than others" [73].

We argue that a key part of this process is recognition of the engineering, environmental, and socioeconomic problems associated with each technology. We stress that the purpose of this review is not to advocate for any particular energy technology, but rather to provide the reader with a greater awareness of the pros and cons of each of the main technologies and energy policies that are currently being promoted. In order to identify these key energy technologies and policies, we have taken advantage of the detailed analysis carried out by the Climate Policy Initiative (https:

//www.climatepolicyinitiative.org/) in a series of annual/biennial "Global Landscape of Climate Finance" reports which have estimated the breakdown of the total global climate change expenditure for each year from 2010/2011 [74] to 2018 [75].

We have compiled the data for each year from these reports in Figure 1 and Table 1. We note that the Climate Policy Initiative also carried out an estimate for 2009/2010 in an early report [76], but the authors advise that they significantly modified their methodology for subsequent reports, and so we have not included these earlier estimates in our analysis. According to its website, the Climate Policy Initiative is a climate policy think tank that "was founded in 2009 to support nations building low-carbon economies to develop and implement effective climate, energy, and land use policies". In their reports, they explicitly acknowledge that their calculations likely underestimate the annual global expenditure, "due to methodological issues related to data coverage and data limitations, particularly domestic government expenditures on climate finance and private investments in energy efficiency, transport, land use, and adaptation" (Buchner et al. 2019, p8) [75]. Nonetheless, they appear to offer the most comprehensive estimates available at the time of writing. Therefore, we believe they offer a useful relative breakdown of global climate change expenditure over the period 2011–2018.

**Table 1.** Breakdown of global climate change expenditure during 2011–2018. Data from the Climate Policy Initiative's Global Landscape of Climate Finance annual and biennial reports, accessed from https://climatepolicyinitiative.org/.

| Sector | Total Expenditure, 2011–2018 (8-Year Period) | Average Annual Expenditure | Percentage of Total Expenditure |
|---|---|---|---|
| Solar | US$ 1220 billion | US$ 152 billion | 33% |
| Wind | US$ 810 billion | US$ 101 billion | 22% |
| *Biomass & waste** | *US$ 75 billion* | *US$ 9 billion* | 2% |
| *Hydroelectricity** | *US$ 75 billion* | *US$ 9 billion* | 2% |
| *Biofuels** | *US$ 25 billion* | *US$ 3 billion* | 1% |
| *All other renewables** | *US$ 170 billion* | *US$ 21 billion* | 5% |
| Sustainable transport | US$ 375 billion | US$ 47 billion | 10% |
| Energy efficiency | US$ 250 billion | US$ 31 billion | 7% |
| Other climate mitigation policies | US$ 430 billion | US$ 54 billion | 12% |
| Climate adaptation policies | US$ 190 billion | US$ 24 billion | 5% |
| Dual benefits | US$ 40 billion | US$ 5 billion | 1% |
| *Total* | *US$ 3660 billion* | *US$ 458 billion* | 100% |

* As solar and wind comprise the bulk of total renewables, the breakdown of "other renewables" is not given in all reports, so we estimated the items marked using figures provided in the 2012 and 2013 reports.

Despite this expenditure totaling US$3660 billion over 8 years, global carbon dioxide ($CO_2$) emissions have continued rising throughout this period (Figure 2). This gives occasion to scrutinize expenditures to consider whether the current path holds promise of success. One explanation could be that the total expenditure is still too low, and indeed Buchner et al. (2019) argue that annual global expenditure would need to increase to US$1.6–3.8 trillion in order to substantially reduce $CO_2$ emissions [75]. However, Figure 1 and Table 1 show that 55% of the expenditure over this period has been on solar and wind projects, with a further 10% on sustainable transport projects and 7% on energy efficiency. That is, most of the expenditure has gone on the types of policies prioritized by the "zero-carbon" proposals which have been heavily criticized above.

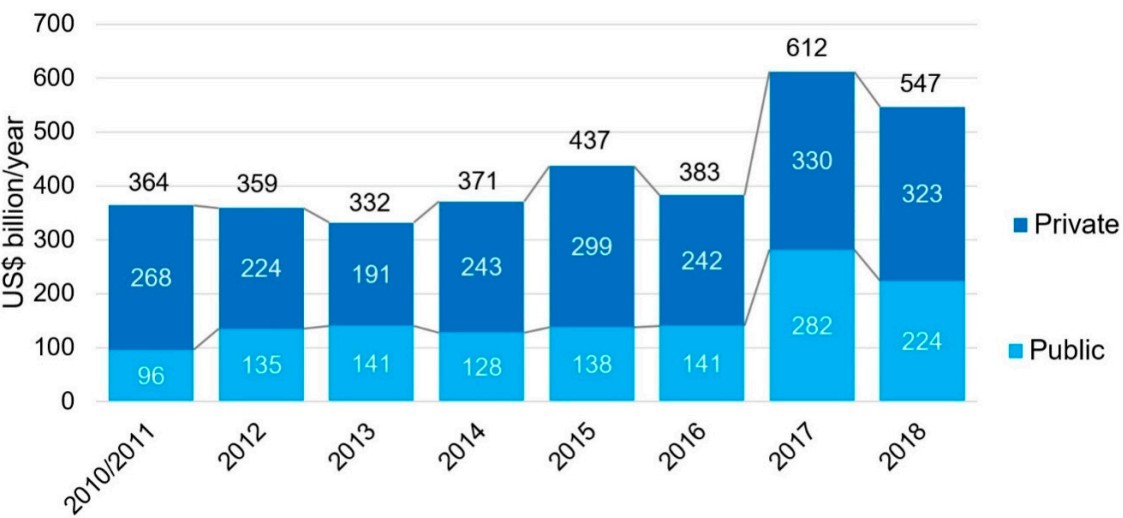

## (a) Total global climate change expenditure (US$ billion/year)

## (b) Average global climate change expenditure (2011-2018)

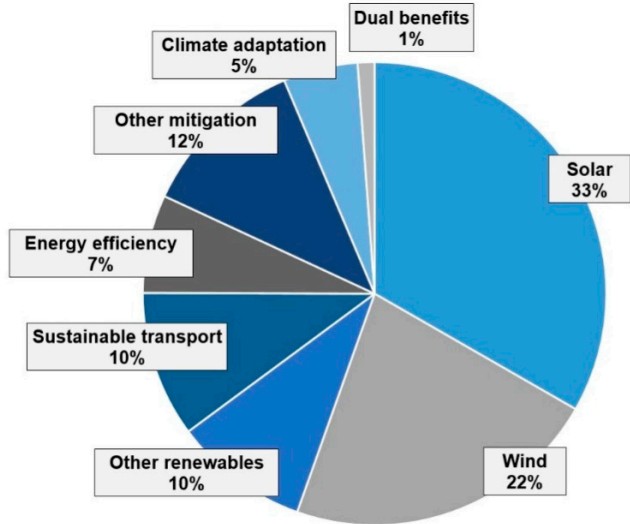

Source: Climate Policy Initiative's
Global Landscape of Climate Finance reports

**Figure 1.** Breakdown of total global climate change expenditure over the period 2011–2018. Data from the Climate Policy Initiative's Global Landscape of Climate Finance reports, accessed from https://climatepolicyinitiative.org/, as detailed in Table 1.

With that in mind, we propose to first describe the world's current energy usage (Section 2). Then, we will consider some of the key engineering challenges associated with both the proposed energy transitions and current energy policies (Section 3). In Section 4, we will consider some of the key environmental concerns associated with these policies, while in Section 5 we consider some important socioeconomic concerns. In Section 6, we summarize the pros and cons of all the main energy sources—both those considered in Figure 1 and Table 1, and those not. In Section 7, we offer some recommendations for how to interpret these conflicting pros and cons.

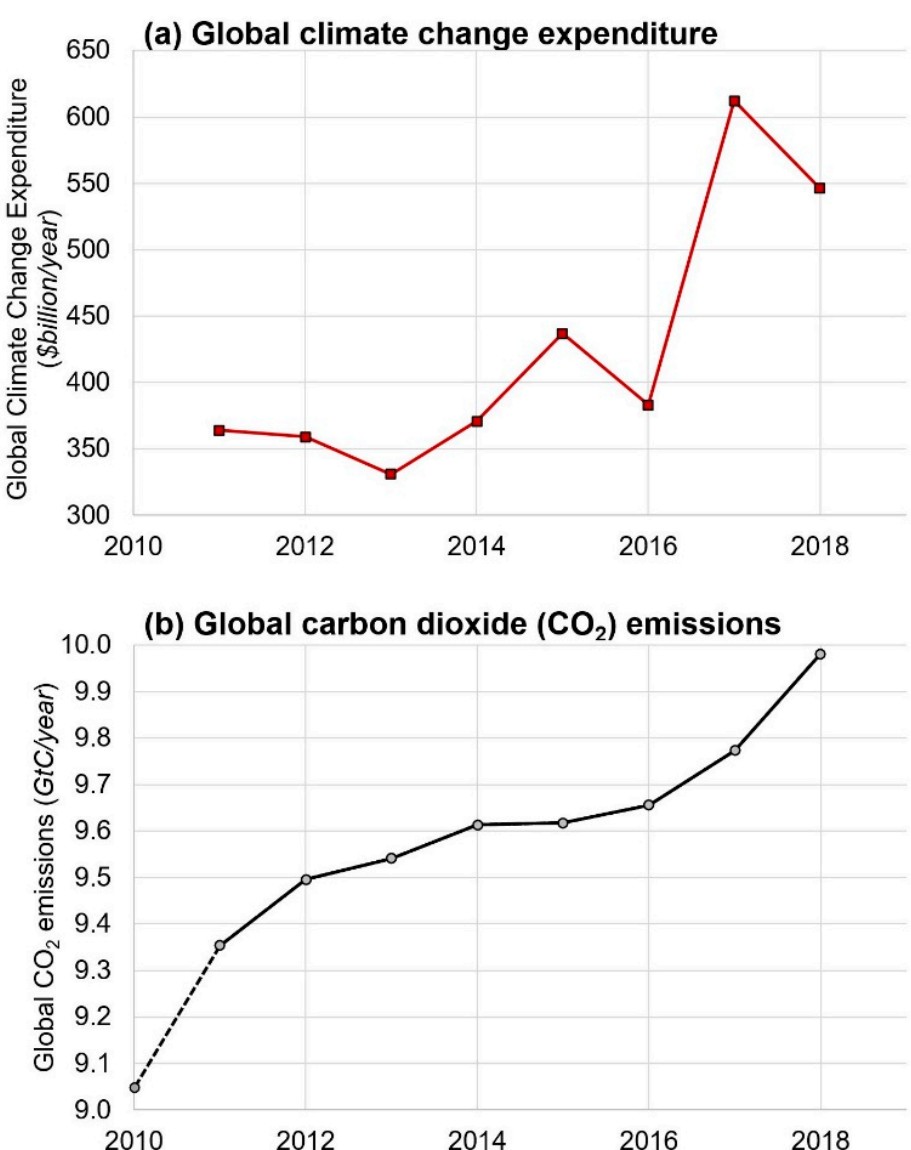

**Figure 2.** Historic trends in global $CO_2$ emissions and global climate change expenditure over the 2011–2018 period. (**a**) Expenditure figures are as in Figure 1. Data from the Climate Policy Initiative's Global Landscape of Climate Finance annual and biennial reports, accessed from https://climatepolicyinitiative.org/. (**b**) Global $CO_2$ emissions. Data from Boden et al. (2018) [4], https://energy.appstate.edu/CDIAC, updated to 2018 by Friedlingstein et al. (2019) [5], https://www.globalcarbonproject.org/.

## 2. Current Energy Policies

In order to evaluate the context of the expenditure policies of Figure 1 and Table 1, it may be helpful to consider the current and historic total world energy consumption. In Figure 3, we present the trends from 2008–2018 as estimated by BP's "Statistical Review of World Energy 2019" [77], and in Figure 4 we show a more detailed breakdown for the most recent year (2018). There are several other similar reports by different groups, and the estimates are broadly similar. We selected this one as it was readily available and was one of the most comprehensive and detailed. Although we recognize a risk that a private energy corporation with a diverse portfolio might have an incentive to misrepresent the relative contribution of various sources of energy to world energy consumption, we note that the US Energy Information Administration (EIA) offers similar estimates for shares of global energy mix in

2018: fossil fuels 80.4%, renewables 15.4%, and nuclear 4.2% [78]. For a comparison of the BP reports to the other world energy reports, see Newell et al. (2019) [79].

We can see from Figure 4 that, as of 2018, the world is still generating most (85%) of its energy from fossil fuels (oil, coal, and gas). Nuclear (4%) and one of the renewables, hydroelectricity (7%), also represent significant slices of the pie. However, wind and solar only represent 3%, and other sources only represent 1%. That said, Figure 3 shows that the large global expenditure on wind and solar projects over the 2011–2018 period has had an effect in that wind and solar represented less than 0.5% of world energy consumption in 2010, so the contribution of wind and solar to the energy mix has increased by 2.5 percentage points over that period.

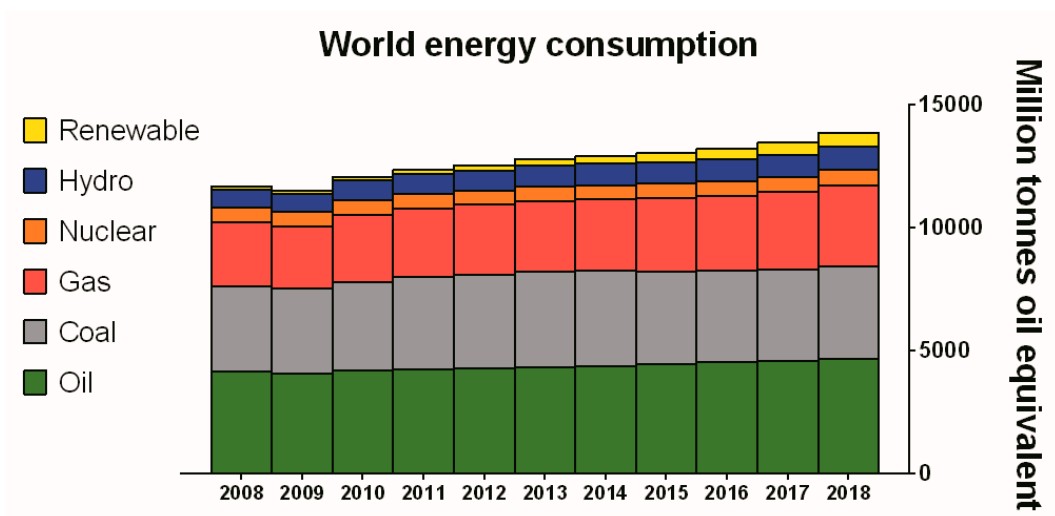

**Figure 3.** World energy consumption by source, ten-year trend (2008–2018). "Renewable" refers to all renewables other than hydroelectricity. Data from BP (2019) [77].

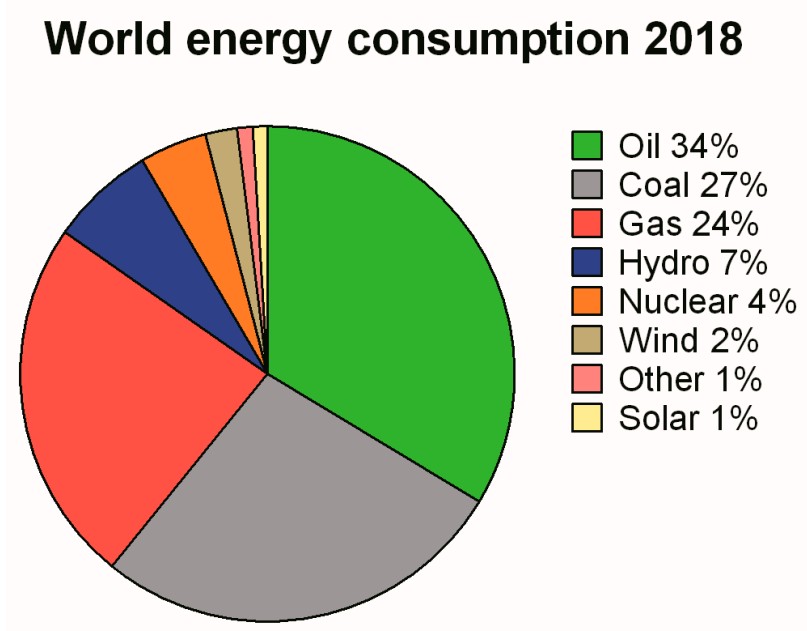

**Figure 4.** World energy consumption by source, 2018. Data from BP (2019) [77].

## 3. Engineering Challenges of the Various Energy Technologies

*3.1. The Intermittency Problem (of Wind, Solar, and Tidal-Generated Electricity)*

Historically, national electricity grids have been almost exclusively powered by "baseload" electricity producers (sometimes called "dispatchable"). While demand for electricity tends to fluctuate at various time scales, mains suppliers are required to provide a steady supply of electricity to meet 'baseload' power needs, defined as "minimum demands based on reasonable expectations of customer requirements" [80]. The following are the most common baseload electricity generation technologies: coal, natural gas, oil, peat, nuclear, hydropower, geothermal, and biomass. As can be seen from Figure 4, these energy sources currently account for more than 96% of the world's energy.

On the other hand, as discussed in the introduction, many of the current proposed energy transitions are heavily reliant on some combination of three "intermittent" (sometimes called "non-dispatchable") electricity generation technologies, i.e., wind, solar, and tidal. Some have even claimed that it is possible (and desirable) to provide 100% of society's energy needs using only renewable energy based mostly on wind and solar [11–18,25–28,32]. Indeed, Jacobson et al. [13–16] advocate for a transition to energy systems that generate 100% of their electricity from wind, water, and sunlight (WWS), i.e., wind and solar supplemented with tidal and hydro. From Figure 1 and Table 1, we can see that 55% of the total global climate change expenditure over the period 2011–2018 has been spent on two of these technologies, i.e., solar and wind. Therefore, it is worth considering the implications of "the intermittency problem". We stress that this is not a problem which has applied in the past to electricity grids using exclusively dispatchable power stations.

In contrast to the steady or on-demand production of baseload power stations, the intermittent technologies only provide energy on an intermittent basis, i.e., only when the wind blows (for wind) or only when the sun shines (for solar) or depending on the tides (for tidal). However, the consumption of electricity by the consumers does not follow these production times. This leads to mismatches between the supply and demand of electricity that become increasingly problematic the greater the amount of intermittent electricity generators connected to the grid. At some times, too much electricity is generated and needs to be "curtailed", i.e., dumped or diminished, while at other times too little is generated, leading to blackouts.

A steady supply of energy on a 24-h basis is indispensable to the safe and reliable operation of systems such as water treatment plants, hospitals, domestic heating/air-conditioning systems, manufacturing plants, and mass transit systems. In addition, ready availability of energy is a prerequisite for the operation of emergency services, e.g., medical resuscitation equipment. From the perspective of the householder, reliable energy is necessary to keep a refrigerator running round the clock, and it must be available at the flick of a switch to provide lighting as needed at any time of night. Without it, the householder is likely to suffer food wastage from refrigeration failures, accidents for want of light, and loss of temperature control due to failure of heating or air-conditioning [81].

Peak electricity demand is subject to a range of uncertainties, including population growth, changing technologies, economic conditions, prevailing weather conditions, and random variation in individual usage. It also follows patterns of variation by time of day, day of the week, season of the year, and public holidays [82]. The pattern of electricity use in an individual household is highly dependent upon the activities of the occupants and their associated use of electrical appliances [83]. Figure 5 illustrates the variations in energy consumption for an individual household by time of day or night. Note in particular the very low consumption by night, with occasional spikes as appliances are turned on for short periods, and generally higher consumption during the day, with brief spikes of even higher consumption at irregular intervals. Although these demand patterns are quite noisy on an individual basis, when averaged over an entire country, the national demand is relatively predictable.

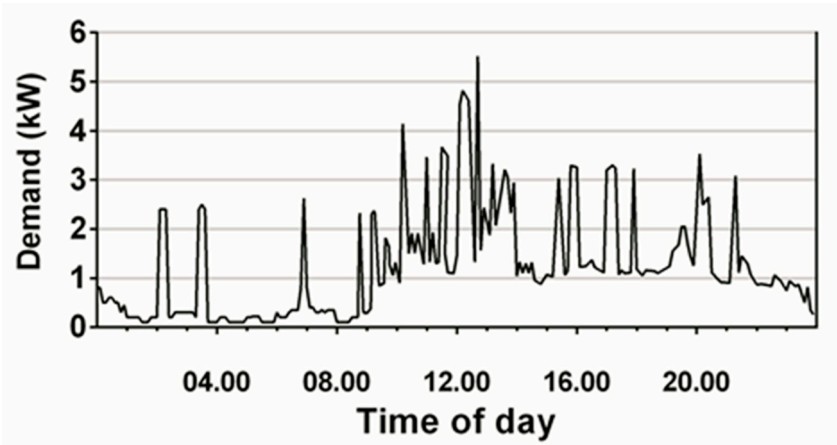

**Figure 5.** Sample measured daily demand profile for domestic electricity, in one-minute intervals, from a dwelling in the East Midlands, UK. (Adapted from Figure 5 of Richardson et al. (2010) [83]).

For example, the blue curve in Figure 6 shows weekly and seasonal variation in electricity demand in the Republic of Ireland throughout a full year (2013). Note the regular pattern of high demand on weekdays and lower demand at the weekend, with seasonal variation imposed over this. Minimum demand was approximately 2500 MW in summer and approximately 3000 MW in the winter months of 2013. Compare this with the red curve in Figure 6 which shows the fluctuating character of wind energy generation in the Republic of Ireland for the same year (2013). Wind turbines produced more than 1000 MW on 14% of the days and less than 100 MW on 10% of the days. Note in particular, the low levels of electricity generated in the last week of February, the second week of July and the last few days of November.

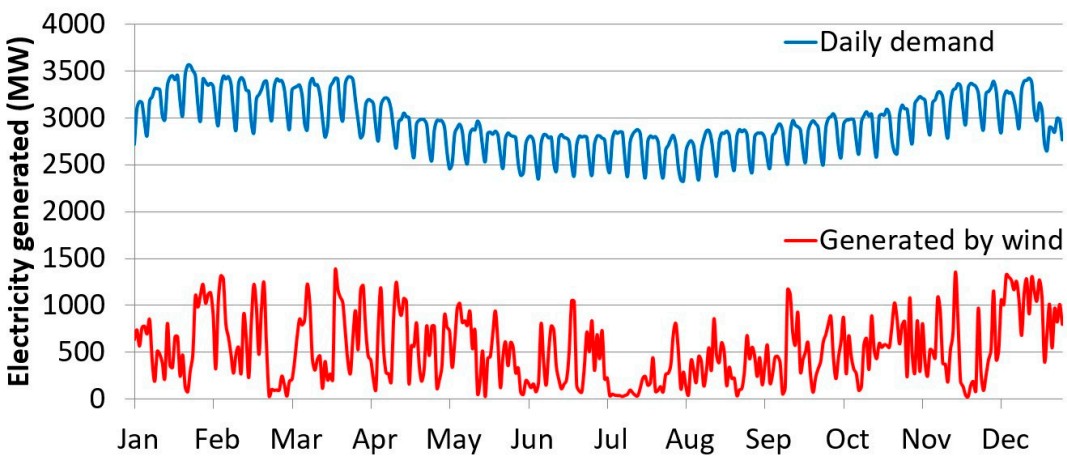

**Figure 6.** Annual variation in daily electricity demand (blue) and electricity generated by wind (red) Republic of Ireland, 2013. (Data from: time series downloaded from http://www.eirgrid.com/ in January 2014.).

Solar power is a bit more predictable in that most of the intermittency occurs from the day/night cycle, although variability in cloud cover creates an additional chaotic component. However, for mid-to-high latitudes, a major problem arises because of the seasonal changes in total sunlight between winter and summer. For instance, as can be seen from Figure 7, the available solar energy in Ireland varies by a factor of ten between December (0.46 kWh/day) and June (4.66 kWh/day). Furthermore, the length of day in December (~8 h) is only half that in June (~16 h).

In addition to the variability at the scale of days and weeks, outlined above, local climates also vary from year-to-year, and climate change can introduce long-term climatic trends which could alter the expected electricity generation even further.

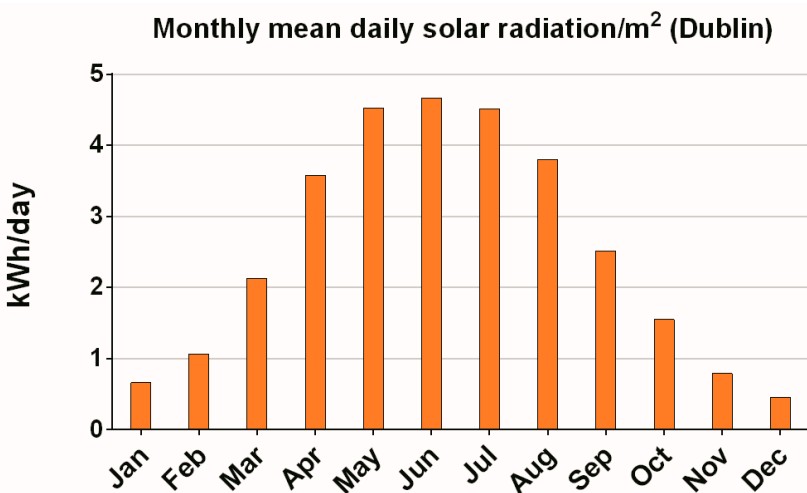

**Figure 7.** Monthly mean daily solar radiation per m2 on a horizontal surface at Dublin Airport, 1976–1984 (Adapted from Table 36 of Rohan (1986).) [84].

When the percentage of intermittent electricity is relatively low, the remaining baseload generators can reduce some of the problem by ramping up or down production in response to the intermittency. However, this raises several problems. First, the electricity grid now requires a much higher total capacity because it still needs to have near 100% capacity as before in order to be on standby to provide electricity when the intermittent generators are not in operation. Moreover, the need to switch back and forth is very wasteful (as well as reducing energy efficiency). Carnegie et al. (2013) note that balancing electricity generation and load using traditional baseload power plants (fossil fuel, hydroelectric, and nuclear) can be "costly in terms of capital life expectancy and operational inefficiencies. Frequent adjustment of generation output increases the wear and tear on generators, reduces their expected lifetimes and increases maintenance expenses. This operational scheme also results in both cost and productivity inefficiencies" [85]. Meanwhile, if a gas turbine is designed for baseload power generation but subsequently used for load balancing in conjunction with intermittent energy sources, the constant acceleration and deceleration of the shaft severely shortens the lifespan of the turbine [20].

Various solutions have been proposed to solve the problem of intermittency. A utility company can attempt to balance power supply from intermittent sources over very large areas by constructing an extended transmission network and coordinating energy production from the different sources, e.g., solar and wind installations, but this requires significant capital investment [50]. This approach increases the minimum capacity needed and can give rise to "bottlenecks", i.e., delays in energy transmission to large centers of demand, often distant from sites of energy generation [80]. Moreover, weather patterns tend to affect quite large geographic areas at similar times, e.g., if it is unusually windy or calm in France, it is likely to be the same in Germany. Other options include "demand-side management, electricity storage, and enhanced coordination or forecasting of power plants" [80].

Some researchers have argued that, in principle, the intermittency problem could be reduced through energy storage [48,49]. That is, when a wind farm or solar farm is producing too much for demand, it could store the excess electricity using some form of energy storage technology. Then, when demand increases above supply, this stored energy could be returned. However, others have pointed out that the storage capacity required would be unrealistically enormous, and satisfactory resolutions to this problem have not yet been demonstrated with available technologies [19,21–24,45].

For instance, van Kooten et al. (2020) note that, although Tesla have recently "built what is considered to be a gigantic, 100-MW (MW)/129-MW-hour (MWh) capacity battery in South Australia to address blackouts resulting from renewable energy intermittency" [24], they calculate that, if the state of Alberta (Canada) were to rely solely on intermittent electricity sources for generating electricity, it would need the equivalent of 100 such batteries. Shaner et al. (2018) calculated that even to meet the lesser target of 80% wind/solar for the United States would require enormous and unprecedented

infrastructural investments. The exact requirements would depend on whether the grid was mostly wind or mostly solar. For a solar-heavy grid, enough energy storage would be required to overcome the daily solar cycle, i.e., 12 h worth of energy storage (~5.4 TW h). For a wind-heavy grid, it would require a continental-scale electricity transmission network "to exploit the geographic diversity of wind" [22]. They further calculated that, "to reliably meet 100% of total annual electricity demand, seasonal cycles and unpredictable weather events require several weeks' worth of energy storage and/or the installation of much more capacity of solar and wind power than is routinely necessary to meet peak demand [ . . . ] Today this would be very costly" [22]. More generally, Heard et al. (2017) criticize the "near-total lack of historical evidence for the technical feasibility of 100% renewable-electricity systems operating at regional or larger scales. The only industrialized nation today with electricity from 100% renewable sources is Iceland, thanks to a unique endowment of shallow geothermal aquifers, abundant hydropower, and a population of only 0.3 million people" [21].

### 3.2. The Power Density Problem

In comparing energy options, it is useful to calculate how much land is required for each energy technology and how much energy this can supply. This calculation is known as the power density and is defined as the energy generation rate per time per unit ground area (expressed as $W/m^2$). Smil (2005) points out that the proposed energy transition to renewables calls for "an order of magnitude larger displacement of dominant resources than during the last major energy transition" [86], i.e., the transition from burning biomass to fossil fuels. The inherently low efficiency of photosynthesis means that biomass harvests do not surpass 1 $W/m^2$, while most fossil fuel extraction proceeds at rates exceeding 1000 $W/m^2$. Replacing crude oil-derived fuels by less energy-dense biofuels would also require commonly 1000-fold and often 10,000-fold larger areas under crops than the land claimed by oilfield infrastructures.

Figure 8 compares the average power densities for most of the main energy sources as estimated by Zalk et al. (2018) [87]. It can be seen that the power densities of nonrenewable energy (non-RE) sources are up to three orders of magnitude greater than those of renewable energy (RE) sources. In other words, they produce about a thousand times as much power for any given land surface area. Natural gas yields the highest median power density by far. Of the renewable energy sources, solar energy yields the highest median power density, but is still orders of magnitude lower than either nuclear or the fossil fuels. However, the lowest of all nine of the technologies is biomass.

In Section 4.4, we will discuss the negative implications of the increased reliance on low power density sources (particularly biomass/biofuels) have for biodiversity, including increases in deforestation rates. Moreover, in Section 5, we will discuss some of the associated negative socioeconomic effects. However, in this subsection, we stress the simple logistical problems that this implies from an engineering perspective. Indeed, arguably, this is the most challenging of the engineering problems we discuss in this paper in terms of a proposed energy transition from a society that currently gets 89% of its energy from fossil fuels and nuclear (85% from fossil fuels alone) to one that relies mostly on the renewable energy sources (see Figures 3 and 4). It is true that before the Industrial Revolution, society derived most of its energy from low power density renewable technologies similar to those in Figure 8 (much of the energy usage came from human or animal labor, indirectly fueled by biomass, i.e., food). However, the world's population in 1800 was only ~1 billion compared to ~7.8 billion today, and most of those ~7.8 billion people would probably not be satisfied with a return to pre-industrial standards of living [7–9].

Therefore, as high power density energy sources are replaced with low power density energy sources, the land area required to be set aside for energy production dramatically increases. As a result, the average energy footprint per capita (and as will be discussed in Section 4.4, the corresponding ecological footprint) [88] will rise accordingly. This should be particularly concerning for those who believe we are currently "overpopulated" (see Section 3.3.1). It should also be concerning to readers who believe that societies in developing nations that currently have very low energy footprints

(including the ~1 billion people without access to electricity) should be encouraged to increase their energy footprint (see Section 5).

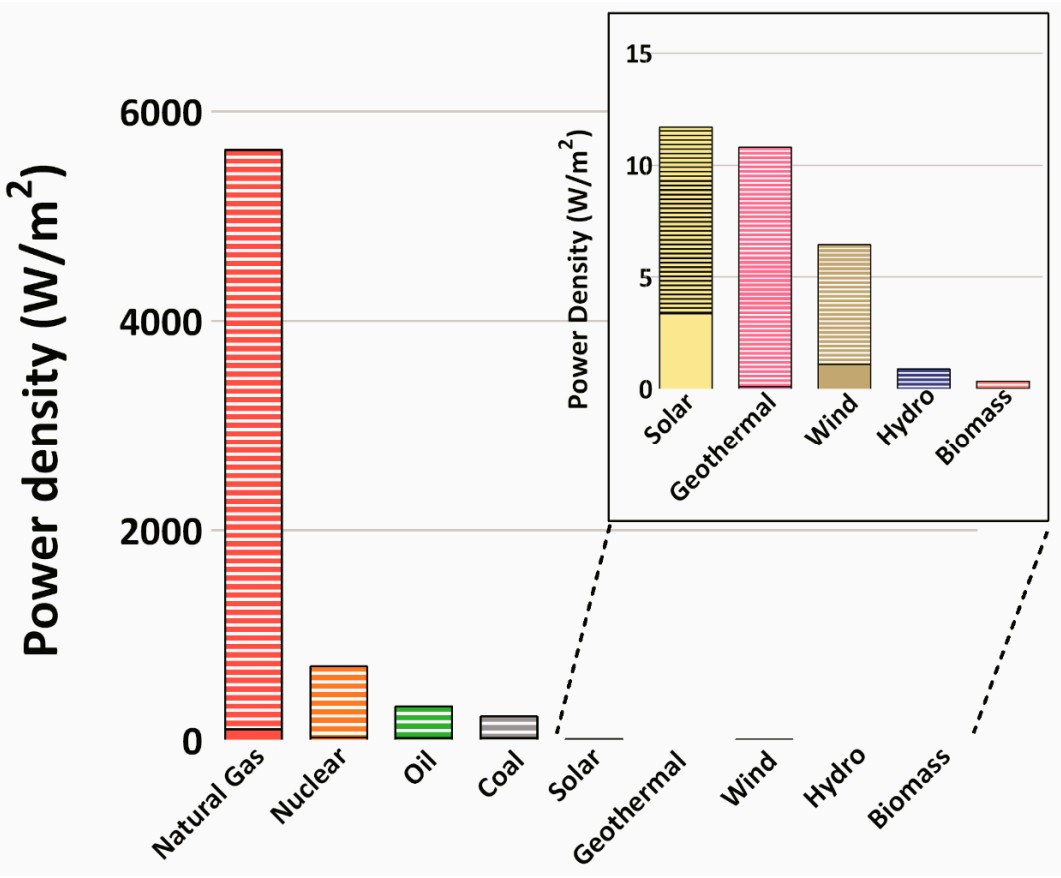

**Figure 8.** Power densities for most of the main electricity generation sources. Hatched areas indicate values between minimum and maximum estimates. (Adapted from van Zalk (2018) [87]).

### 3.3. The Limited Resources Problem

#### 3.3.1. The Neo-Malthusian Debate; "Sustainable" Versus "Renewable"

Malthus (1798) warned that the growth in the world's population (then ~1 billion) would quickly lead to catastrophic consequences as he argued that food production could not keep pace with population growth. He concluded that unless birth rates decreased significantly and urgently this would lead to famine and devastation [89]. Although the population is currently ~7.8 billion, more than two centuries later, suggesting that his predictions were badly flawed, his logic was compelling to many at the time.

Equivalent logic has led many researchers to make updated predictions along similar lines over the years since [90–92]. Essentially, the logic suggests that, all else being equal, if per capita consumption of some limited resource is constant or increasing, but the population continues to increase, at some point demand will outstrip supply. If society is reliant on this resource, this could potentially have devastating effects. Analogies with the human population are sometimes made with ecological systems that go through boom/bust cycles, e.g., bacterial growth on a Petri dish with nutrient agar can be rapid until all of the nutrients are consumed, at which stage the colony can completely collapse [90,92]. Due to the similarity of the logic to that underlying Malthus' predictions, this is often referred to as the "neo-Malthusian argument".

Intuitively, the logic behind the neo-Malthusian argument is initially compelling. However, critics invariably point to the fact that, empirically, the observed trends are often contrary to the

trends predicted by the neo-Malthusian theories [44,93–95]. Critics note that a key weakness in the neo-Malthusian argument is the assumption that society does not modify its usage of a resource in response to the supply/demand ratios. Moreover, that humans can invent new approaches and technologies. A common pithy counter to the neo-Malthusian argument is to note that "the Stone Age didn't end because we ran out of stone". Simon, noting that human ingenuity distinguishes us from bacteria on a Petri dish, went so far as to refer to humanity as the "ultimate resource" [93].

Some critics of the neo-Malthusian argument have even argued that the underlying logic is so flawed as to propose the opposite, i.e., that humanity can keep growing for the foreseeable future, provided we allow our citizens the opportunity to avail of their ingenuity. This has led some to refer to criticism of neo-Malthusian arguments as "cornucopianism" [96].

We will not comment here on which side in this debate is closest to the truth, but merely note, first, that the debate seems to have been recurring in slightly different forms for more than two centuries now [94]. Second, despite the fact that neo-Malthusians often express their predictions of future trends with remarkable confidence [90–92], these predictions are frequently found retrospectively to have been opposite to reality [95], indeed the world's population is now nearly 8 times as large as during Malthus' original predictions.

This has important implications for what we regard as "sustainable". Societies differ in their conceptualizations of sustainability, as defined by numerous disciplines and applied to a variety of contexts. These range from the concept of maximum sustainable yield in forestry and fisheries management to the vision of a sustainable society with a steady-state economy. Brown et al. (1987) [97] proposed that indefinite human survival on a global scale requires certain basic support systems, "which can be maintained only with a healthy environment and a stable human population", which corresponds with the neo-Malthusian perspective in explicitly incorporating human population trends into the mix. However, given that the "energy footprint", "ecological footprint", etc. of individuals can vary over time and from region to region [88,98], we argue that it is meaningless to define an arbitrary "ideal" population size above which the world becomes "overpopulated". Instead, we suggest we should avoid defining "sustainability" explicitly or implicitly in terms of population trends (i.e., the neo-Malthusian paradigm). For instance, Gomiero (2015) argues that, "In order to be termed sustainable, the use of an energy source should be technically feasible, economically affordable, environmentally and socially viable, considering society as a whole" [99]. This definition does not explicitly depend on population trends, although clearly the size of the population is an important factor to consider.

The literature offers several definitions of renewable energy (RE) supplies. For example, a special report from the IPCC on renewable energy sources and climate change mitigation offers the following. "Renewable energy is any form of energy from solar, geophysical or biological sources that is replenished by natural processes at a rate that equals or exceeds its rate of use" [100]. Verbruggen et al. (2010) point out that this definition can be refined, for example, by adding the notion that some renewable sources can be exhausted by overexploitation. Conversely, they caution, "qualifying the various renewable energy supplies for measuring their degree of sustainability is an unsolved issue" [101]. Moreover, Acosta (2013) cautions that intensive resource extraction may blur the distinction between renewable and non-renewable sources of energy: "Because of the huge scale of extraction, many 'renewable' resources, such as forests or soil fertility, are becoming non-renewable. This is because the resource is depleted when the rate of extraction is much higher than the rate at which the environment is able to renew the resource. Thus, at the current pace of extraction, the problems of non-renewable natural resources may affect all resources equally" [102].

Thus, we suggest that there may be unsustainable exploitation of a renewable energy source (e.g., clearance of a forest for the manufacture of wood pellets) and, conversely, there may be sustainable exploitation of a non-renewable source (e.g., scheduled management of a finite reserve of natural gas to last over a planning period of 10 or 20 years). Therefore, the sustainability of a resource is not merely a question of whether it is finite or renewable, but of how it is managed with regard to its lifespan or life cycle. For instance, in history, we can see that the use of fossil fuels (a non-renewable

resource) for energy arose precisely because the continued burning of wood (a renewable resource) was unsustainable. The use of coal was found to be more sustainable than continued deforestation [8].

### 3.3.2. "Peak Oil", "Peak Gas", and "Peak Coal"

Since M.K. Hubbert first coined the term "peak oil" in the 1950s, the argument that resources such as oil are finite has been a recurring motif in energy policy discussions [103–107]. The argument is that, if society relies too heavily on oil (or gas), then there may be catastrophic consequences if demand suddenly outstrips supply, because we have reached "peak oil" or "peak gas". Readers may note this argument overlaps with the neo-Malthusian arguments described above. Indeed, concerns about "peak oil" have been particularly prominent in neo-Malthusian analyses since the 1970s [90–92]. However, as we noted above with regards to the neo-Malthusian debate, predictions of imminent "peak oil", "peak gas", and "peak coal" are continually being revised forwards as time progresses.

Indeed, Lior (2008) noted, "An interesting global phenomenon is that despite the rise in consumption of fossil fuels, the quantities of proven reserves rises with time too, where the resources/production (R/P) ratio has remained nearly constant for decades at R/P = 40 for oil, 60 for gas and about 150 for coal" [108]. This implied that there was at least 40 years (oil), 60 years (gas), and 150 years (coal) of reserves at 2006 consumption rates, but that we should not be surprised if several decades from now, the future predictions for peak oil, gas, and coal will have moved forward in tandem. Shafiee and Topal (2009) disputed whether this empirical observation is a reliable assumption for projecting forward and estimated that the reserves would only last 35 years (oil), 37 years (gas), and 107 years (coal) at 2006 consumption rates [109]. At present, i.e., 10 years later, BP (2019) estimate the total world reserves-to-production ratios (R/P) are 50 years (oil), 51 years (gas), and 132 years (coal) at 2019 consumption rates [77].

We will not attempt to resolve these conflicting estimates in this paper. Nor are we suggesting that coal, oil, and gas should be assumed to be "inexhaustible" resources (although we refer interested readers to Kutcherov and Krayushkin (2010) for an intriguing review on the controversial hypothesis that oil and gas may be "abiotic" in origin, implying that possibility [110]). Rather, we suggest that energy policies which are based on specific predictions of the timings of "peak oil/gas/coal" should be treated with considerable caution. For instance, partially on the basis of predictions of "peak oil", expensive liquefied natural gas (LNG) terminals were developed in the U.S. In the first decade of the 2000s, to import major quantities of gas by developers anticipating a looming shortage of supply in the country. Yet, by the time these terminals were in operation, technological advances in hydraulic fracturing ("fracking" for short) had dramatically increased U.S. gas accessible reserves, switching the country to a net exporter of gas [105,107]. It has been suggested that, if the large-scale extraction of gas from underwater methane hydrate reserves becomes economically viable in the future, even greater increases in "gas reserves" would occur [110,111].

Therefore, there is considerable uncertainty over when exactly we should expect "peak" oil, gas, or coal, but certainly there seem to be enough known reserves of all three of these for the next few decades at least. This has led to conflicting perspectives from researchers concerned about anthropogenic global warming from $CO_2$ emissions over whether we should continue to use fossil fuels as long as they are readily available on the basis that there is not much left [112] or actively campaign to keep fossil fuels "unused" on the basis that there is too much left [113].

### 3.3.3. The Mineral Scarcity Problem

Because of the 10-fold increase in quantities of minerals required by green technologies relative to those driven by hydrocarbons, Mills (2020) cautions that any significant expansion in green energy will create "an unprecedented increase in global mining", which would radically exacerbate environmental and labor challenges in emerging markets, and dramatically increase the vulnerability of America's energy supply chain [45]. Capellán-Pérez et al. (2019) underscore the concern that the extraction of the minerals required for the proposed transition to renewable energies is likely to intensify current

socio-environmental conflicts associated with resource extraction [114]. As we will outline in the following section, this gives rise to concern regarding potential uncertainty of supply. In contrast to the concerns about hydrocarbon peaks outlined above, projected mineral requirements seem likely to exceed current reserves within the very short time frame to the year 2030. This concern appears particularly pressing with regard to e-vehicles, which we discuss next, followed by related concerns regarding solar and wind energy.

The projected production of electric vehicles (EVs) to replace vehicles powered by fossil fuels requires the consumption of a new range of metals, as outlined in a letter from a group of geologists and other earth scientists, led by Professor Richard Herrington, Head of Earth Sciences at the Natural History Museum [58], to the Committee on Climate Change in London who had recommended increasing the percentage of the UK's cars that are electric or hybrid from 0.2% in 2017 to 100% by 2050.

Herrington et al. warn that in order to replace the UK's fleet of cars (currently 31.5 million) entirely with EVs, it would require "just under two times the total annual world cobalt production, nearly the entire world production of neodymium, three quarters the world's lithium production and at least half of the world's copper production during 2018 [ . . . ] If we are to extrapolate this analysis to the currently projected estimate of 2 billion cars worldwide, based on 2018 figures, annual production would have to increase for neodymium and dysprosium by 70%, copper output would need to more than double and cobalt output would need to increase at least three and a half times for the entire period from now until 2050 to satisfy the demand" [58]. They further note that this proposed transition for the UK would also lead to a 20% increase in electricity usage for the country, due to the extra power generated needed for recharging the vehicles.

In a spatial analysis of lithium availability, Narins (2017) [115] describes a "contemporary scramble" for this mineral for use in e-vehicle batteries that is "full of contradictions that can be best understood as a global lithium consumption–production imbalance." While Bolivia is the country with the largest known reserves of the mineral, it is not among the world's largest producers and suffers from "undeveloped infrastructure, fickle regulatory environment and uncertainties surrounding the security of mining investments". Although he does not think that the rise of the industry will "ultimately" be constrained by the availability of lithium, because new reserves and methods of extraction are under development, and because it may be possible to use substitutes such as zinc, he signals the current situation that "quality of lithium and price are constricting factors that continue to bring uncertainty to the growth and rate of expansion of the global electric car industry" [115].

Even under its modest "New Policies Scenario", the International Energy Agency's projections to the year 2030 [116] indicate that cobalt and lithium reserves are inadequate to meet EV needs (Figure 9).

Modeling on the assumption of a shift to 100% renewable electricity by the year 2050, with lithium-ion batteries accounting for approximately 6% of energy storage and 55% of energy for road transport being accounted for by electric vehicles, Giurco et al. (2019) [117] consider that the cumulative demand for both cobalt and lithium is likely to exceed current reserves unless recycling rates are improved. They consider that the annual demand for cobalt for EVs and storage could exceed current production rates by around 2023, and that the annual demand for lithium could exceed current production rates by around 2022. Although they consider that high recycling rates can keep cumulative demand for cobalt and lithium below current resource levels, they caution that there is likely to be a delay before recycling can offset demand until there are enough batteries reaching end of life to be collected and recycled.

From extensive field research, including expert interviews, community interviews with miners and traders, and observation at 21 mines and nine affiliated mining sites, Sovacool (2019) [118] documented displacements of indigenous communities, unsafe work environments, child labor, and violence against women in communities near cobalt mines. Because most of the world's cobalt is produced in the Democratic Republic of Congo, the major increases in demand arising from global interest in EVs have created a rise in the number of local "artisanal" mines extracting cobalt. Several journalists have

warned that these are often poorly regulated and sometimes involve the use of child labor [119,120]. These socioenvironmental issues give rise to further concern regarding security of supply.

Capellán-Pérez et al. (2019) identify the technologies most vulnerable to mineral scarcity to be solar PV technologies (tellurium, indium, silver, and manganese), solar CSP (silver and manganese), and Li batteries (lithium and manganese) [114]. The transition to alternative technologies will also intensify global copper demand by requiring 10–25% of current global reserves and 5–10% of current global resources. The authors report that "other studies considering a full transition to 100% RES and considering the material requirements for transportation of electricity reach higher levels, e.g., 60–70% of estimated current reserves".

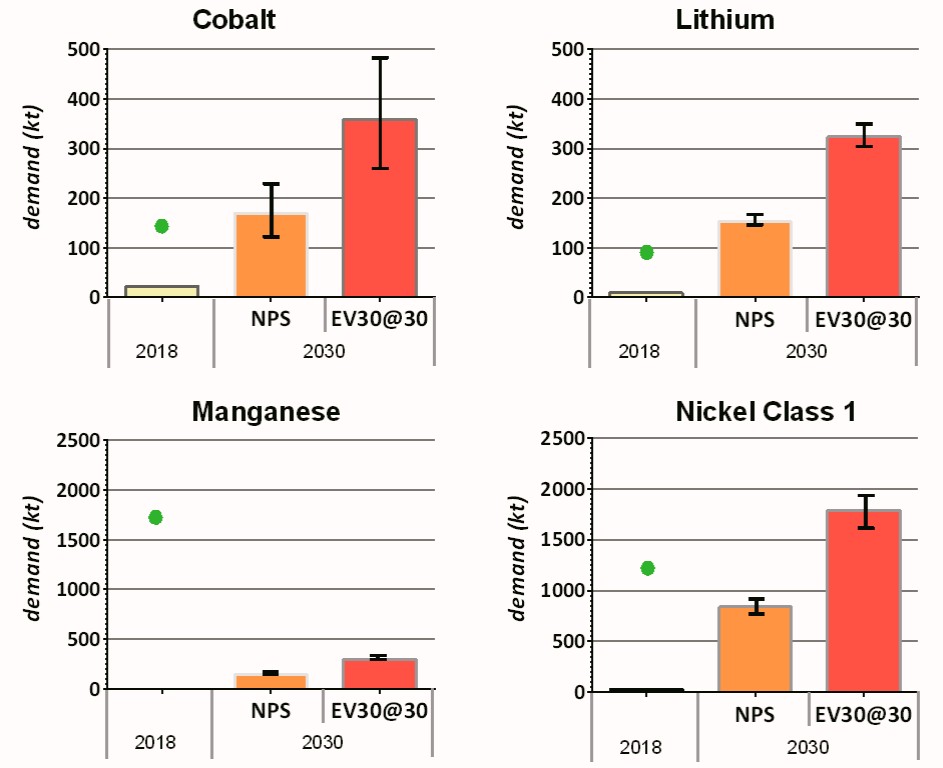

**Figure 9.** Increased annual demand for materials for batteries from deployment of electric vehicles by scenario, 2018–2030. Green dots indicate current supply. NPS = New Policies Scenario. EV30@30 = 30% sales share for EVs by 2030. (Adapted from Figure 7 of IEA, 2019) [116].

Modeling on the assumption of a shift to 100% renewable electricity by the year 2050, with solar PV accounting for more than one-third of capacity and the remainder being generated by wind and other renewables, Giurco et al. (2019) calculate that to generate one-third of the world's energy from solar power by 2050, this would require ~50% of the current reserves of silver [117]. They consider that increasing efficiency of material use has the greatest potential to offset the demand for metals for solar PV, while recycling has less potential because of the long lifespan of solar PV metals and their lower potential for recycling. They also caution that declining ore grades may have a significant influence on energy consumption in the mining sector, associated with polymetallic ore processing and the mining of deeper ore bodies. They note that, although silver has an overall recycling rate of 30–50% almost no recycling of silver from PV panels occurs, because most recycling of PV panels focuses on recycling the glass, aluminum, and copper.

Several types of wind turbine, such as the permanent magnet synchronous generator (PMSG), require magnets that orient wind turbines into the wind. These magnets contain rare metals such as neodymium (Nd), praseodymium (Pr), terbium (Tb), and dysprosium (Dy) [121]. The estimated demand for Nd is projected to increase from 4000 to 18,000 tons by 2035, and for Dy from 200 to

1200 tons [122]. These values represent a quarter to a half of current world output [122]. There are also concerns over the amount of toxic and radioactive waste generated by these mining activities. Current research is focusing on lowering the dependence on these materials by reducing and recycling [123,124].

The construction of extensive wind and solar energy installations will require large quantities of base metals such as copper, iron and aluminum, which will be unavailable for recycling for the lifetime of the installation, thus exacerbating scarcities (Vidal et al., 2013) [125].

## 4. Environmental Concerns Associated with the Various Energy Technologies

### 4.1. Reducing Greenhouse Gas Emissions

As stated above, the IPCC defines climate mitigation as an intervention to reduce the sources or enhance the sinks of greenhouse gases (GHG). This might be accomplished inter alia by switching from carbon-intensive to less carbon-intensive energy sources.

In Figure 10, we have compiled together estimates of the average GHG emissions from each of the main electricity generation technologies from several sources [126–130]. "Direct emissions" are those arising from power plant operation. "Indirect emissions" (indicated by an "*" in the figure) include all processes and associated emissions except power plant operation, categorized as "upstream" (e.g., oil extraction and refining, coal mining, and fuel transport) or "downstream" (e.g., decommissioning and waste disposal). Note that, while Weisser (2007) [126] considered wind turbines and solar energy operation to be emissions-free, a meta-survey by Nugent and Sovacool (2014) [127] found a range of operational emissions. We do not consider here the potential additional increase in biological $CO_2$ emissions from wind farms, which will be discussed in Section 4.2.4.

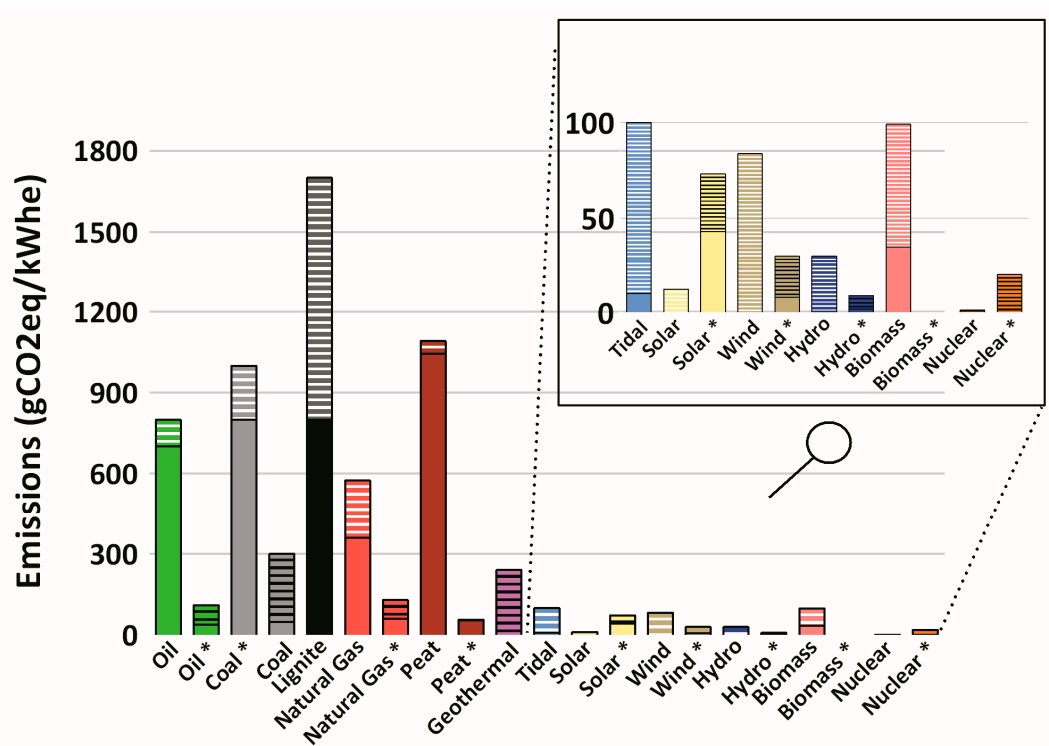

**Figure 10.** Direct and indirect (*) greenhouse gas emissions from electricity supply technologies in grams of $CO_2$-equivalent greenhouse gases per kWh of electricity produced (gCO$_2$eq/kWhe). Hatched areas indicate values between minimum and maximum estimates. Data from various sources, as follows. Most values are from Weisser (2007) [126], but additional estimates for individual technologies are from Nugent and Sovacool (2014) [127]; Eberle et al. (2017) [128]; Paredes et al. (2019) [129]; Murphy et al. (2015) [130].

It should be apparent from Figure 10 why $CO_2$ emissions are highly correlated with fossil fuel usage. The largest $CO_2$ emitters per kWh of electricity are coal (of which lignite is one form that has particularly high emissions), peat, oil, and to a lesser extent natural gas. We include peat here for reference, but we note that it is currently only used in a few countries that have large peat bogs, e.g., Ireland [130].

As we saw in Figure 4, 85% of the world's energy usage in 2018 came from coal, oil, or gas. Therefore, some of the most obvious ways to reduce global $CO_2$ emissions would be to increase the relative percentage of electricity being generated by the technologies that emit less $CO_2$ per kWh. However, unfortunately, one of the main reasons why 85% of the world's energy use still comes from fossil fuels is because they have so many advantages in terms of minimizing the engineering problems discussed in Section 3, as well as dealing with many of the socioeconomic problems discussed in Section 5 [7–9]. At any rate, this is the main rationale behind each of the following strategies for reducing global $CO_2$ emissions from electricity production.

(1) **Coal/oil to gas**. Transitioning from using the higher $CO_2$-emitting coal and oil (and peat) to using more natural gas. This keeps many of the advantages of still using fossil fuels, but significantly reducing total $CO_2$ emissions [131–137]. For instance, de Gouw et al. (2014) calculated that, "Per unit of energy produced, natural gas power plants equipped with combined cycle technology emit on an average 44% of the $CO_2$ compared with coal power plants" [131]. The transition to gas has also been found to significantly reduce air pollution (as we will discuss in Section 4.3.1) [131–134]. Some have argued that such a transition might reduce the motivation to completely abandon fossil fuels [137,138], and others have argued that, if substantial methane leakage is associated with such a transition, the net greenhouse gas emissions may still be high [139]. Nonetheless, it has been argued that in many ways this is the easiest and simplest "short-term" transition to immediately reducing global $CO_2$ emissions (which are still increasing, as can be seen from Figure 2) [41,131–137].

(2) **Carbon capture and storage (CCS)**. In terms of reducing $CO_2$ emissions more completely, the implementation of CCS technology has considerable appeal. Essentially, the emissions from combustion are captured, compressed into a dense fluid, then transported via pipelines and injected into underground storage facilities. This would mean that power plants could continue to use fossil fuels as before, with little or no emissions. On paper, this appears an almost perfect solution for reducing global $CO_2$ emissions, and technology for doing this exists, yet it still has not been implemented on a large enough scale to substantially reduce global emissions [134,140–142]. The main problem is that carbon capture consumes 15–30% of energy from new power plants, and the resultant increase in overall costs currently makes this option economically unviable [142]. Furthermore, storage requires suitable geological sites, such as saline aquifers or abandoned oil fields [142]. Therefore, many have argued that more research and development should be placed into improving CCS technologies to a level where they are economically viable enough for wide-scale implementation [134,140–142].

(3) **Improving energy efficiency**. A different approach to reducing the emissions from electricity generation (and energy usage more broadly) is to reduce the amount of electricity (and energy) used by society, that is, to improve the efficiency of energy usage. As the cost of energy is often a key component of the cost of many activities, it is often (mistakenly) assumed that improving energy efficiencies always makes economic sense too. Clearly, many improvements in energy efficiencies can also make economic sense and/or be emotionally satisfying. However, usually, total energy usage is not the only factor that needs to be considered. Therefore, in a survey of the shipping sector to gauge the implementation of over 30 energy efficiency and $CO_2$ emissions reduction technologies, Rehmatulla et al. (2017) found that, "the measures with high implementation have tended to be those that have small energy efficiency gains at the ship level, and the uptake of $CO_2$-reducing technologies, particularly alternative fuels, is low despite their high potential for reducing $CO_2$ emissions" [143]. Even when energy efficiency measures

might be cost-effective, it has been well-recognized that they are often only slowly adopted—a phenomenon referred to as "the energy efficiency paradox" [144,145].

We can convey the general concepts of the compromises involved in energy efficiency policies by considering the question of how well insulated a house in a mid-to-high-latitude country should be. Historically, many houses were built without much insulation in mind—especially for older houses when internal heating was limited or nonexistent. Therefore, for a relatively small economic outlay, it can be straightforward to reconvert a poorly insulated house into a moderately insulated house. The economic return on investment (ROI) can be substantial, and easily justified. However, after these "low-hanging fruits" have been removed, the ROI on further efforts to improve the insulation falls as the costs and efforts involved tend to increase, while the savings tend to become smaller. At some point, the ROI in increasing insulation may become too low to justify. Moreover, it is often cheaper and easier to incorporate improved insulation techniques and features into a new building than an old building. Therefore, at some point, the ROI involved in "refurbishing" an old building may make less sense than building a highly insulated building from scratch. See MacKay (2009) for a highly informative discussion on these issues [146].

Meanwhile, economists have long debated a separate problem associated with energy efficiency, known as the "rebound effect" [147,148]. It is also sometimes called the "Jevons paradox" [148] after W.S. Jevons suggested in 1865 that this could lead to "peak coal" by the late-19th century (Section 3.3.2). Herring (2006) explains the argument as follows, "that improving energy efficiency lowers the implicit price of energy and hence makes its use more affordable, thus leading to greater use—an effect termed the 'rebound' or 'takeback' effect" [147]. If the improvements are large enough, this may even lead to a "backfire" effect where total energy consumption actually increases as a result of the improved efficiencies. A classic historic example of such an effect is that of the light bulb. A series of energy efficiency improvements in the late 19th century/early 20th century methods of electric lighting were so substantial that it led to a mass market for electric lighting [147,148]. Although these "backfire effects" tend to be rare, many energy efficiency improvements appear to lead to at least some "rebound effect", which partially reduces the expected reduction in emissions [147,148].

(4)  **Increased nuclear usage**. Advocates for nuclear power note that nuclear power generation shares many of the advantages of fossil fuel-generated electricity (see Section 3), without the concerns about peak oil/gas/coal (Section 3.3.2) or emissions [20,21,23,39,41–44,51,52]. Moreover, by studying the energy transitions to nuclear made by both Sweden and France from the 1960s to 1990s, Qvist and Brook (2015) argue that, "if the world built nuclear power at no more than the per capita rate of these exemplar nations during their national expansion, then coal- and gas-fired electricity could be replaced worldwide in less than a decade" [52]. However, even among those calling for the urgent reduction in $CO_2$ emissions, there seems to be considerable public resistance to nuclear energy.

Much of the public concern about nuclear energy seems to be based on concern over these two separate issues: (i) the disposal of radioactive waste and (ii) the risk of accidents [149,150]. With regard to the risk of accidents, Sovacool et al. [151,152] have compiled large databases of major accidents associated with all of the main energy sectors. From analyzing 1085 major accidents related to 11 different energy sectors over the period 1874–2014, Sovacool et al. (2015) found that, although nuclear accidents represented 69.9% of the property damages, they only represented 15.9% of the total accidents and 2.3% of all the associated deaths [151]. The problem of disposal of radioactive waste can be mitigated in at least two ways: third-generation reactors which recycle the fuel, and the use of deep geological repositories (DGRs) [150]. Generation IV reactors currently in development apparently feature further safety, reliability and economic advantages [150].

Another concern is the risk of project overruns in the development of new plants, although this is a concern with any megaproject [153]. In this Special Issue of Energies, Zawalińska et al. (2020) [154] modeled the effects of building a new nuclear power plant in one of four regions in Poland. They

found that the project made sense for one of those regions, but not the other three. That is, new plants should be carefully considered on a case-by-case basis [154] (as with all megaprojects [153]).

(5) **Increased use of hydroelectricity and geothermal**. Hydroelectric power is the leading source of renewable energy worldwide [155], and it has the potential to provide improved energy security by providing abundant, cheap, reliable, and dispatchable energy [73]. Therefore, increasing the percentage of electricity produced from hydropower instead of fossil fuels is one approach to significantly reducing $CO_2$ emissions. However, there are only certain geographical locations where hydroelectric dams could feasibly be built. Therefore, while countries such as Norway can rely predominantly on this option for its electricity, most countries cannot. Also, the associated changes in the landscape are often enormous. As a result, the construction of hydroelectric dams often raises considerable ecological as well as social concerns [73,155,156].

Geothermal energy also has the potential to produce baseload power cheaply, with low levels of $CO_2$ emissions. Geothermal energy can be used either for electricity generation or for direct use, e.g., to provide hot water for industrial and domestic heating [157,158]. Thermal springs in many parts of the world have been steadily producing large amounts of heat and fluid for centuries, and these are renewable as long as a balance is struck between surface discharge and heat/fluid recharge at depth [157]. In some specific geographic regions, it can represent an important resource. For example, it provides 69% of Iceland's primary energy (29% of electricity and 90% of house heating) [158]. However, sites with sustainable high production rates are limited and in many cases not economical [157].

(6) **Fossil fuels to biomass/waste**. The burning of biomass (e.g., wood pellets) and organic waste to produce electricity releases at least as much $CO_2$ as fossil fuels, as fossil fuels are essentially fossilized "biomass" that has been buried underground for millions of years. However, because plants grow by absorbing $CO_2$ from the atmosphere (through photosynthesis), and animals grow by consuming plants, other animals, or decaying matter, it is argued that burning biomass and biofuel is "carbon-neutral". That is, the $CO_2$ released during combustion is balanced by the $CO_2$ absorbed during growth. Therefore, the net $CO_2$ emissions per kWh of electricity are calculated to be much lower than for fossil fuels (Figure 10).

On the other hand, as discussed in Section 3.2, biomass and biofuels have a very low "power density", and as a result, as will be discussed in Section 4.4, there is a lot of concern about the impacts on biodiversity of increasing the use of biomass [51,53–55], and also about the fact that agricultural land that could be used for food or animal feed is being displaced by biomass for fuel [159].

(7) **Increased use of the intermittent renewables (wind/solar/tidal)**. As mentioned in Section 3.1, Jacobson et al. [13–16] advocate for an energy transition towards generating 100% of electricity from wind, water, and sunlight (WWS). The simplicity of this narrative seems to be emotionally compelling among many researchers and writers [11,12,17,18,32] and environmental advocacy groups [25–28]. As a result, the idea that intermittent forms of electricity generation, supplemented with hydro or energy storage systems, could offer a viable alternative to the current systems has become very popular in the public. However, as discussed in the introduction, this idea has been heavily criticized as being physically implausible [8,19–24,26,38,39,41–45], for the reasons outlined in Section 3.

From Figure 1 and Table 1, we can see that global climate change expenditure has predominantly focused on only a few of these strategies. Specifically, 55% of the expenditure has been spent on strategy 7, i.e., increased use of wind and solar; 7% has been spent on strategy 3, i.e., improving energy efficiency; and 2% each on strategies 5 and 6, i.e., hydroelectricity and biomass/waste projects. Strategies 1, 2, and 4 do not appear to be included, and strategy 7 seems to be the main one being pursued, over the period 2011–2018 at least.

Another major source of $CO_2$ emissions comes from the transport sector. In the late 19th century and early 20th century, the rise of coal-powered steam engines revolutionized the transport system

for many countries and the transport sector was heavily powered by coal. However, by the mid-20th century, particularly with the invention of petroleum-powered automobiles, the sector has shifted to being predominantly oil-driven [7]. The aviation and shipping sectors similarly are predominantly oil-fueled. As a result, the transport sector is a major source of $CO_2$ emissions. Therefore, several of the main strategies for reducing $CO_2$ emissions have focused on the transport sector:

(1) **Biofuels**. The rationale for this strategy is equivalent to Strategy (6) for the electricity sector, i.e., to reduce the amount of fossil fuel petroleum that is used by adding biofuels to the diesel or petroleum fuel used by vehicles. As for biomass, it is argued that these fuels are "carbon-neutral" because $CO_2$ is absorbed from the atmosphere as they are being grown. However, as for biomass, biofuels also have a very low power density, so these likewise raise concern about impacts on biodiversity [51,53–55], and about competition with the cultivation of food [159]. There is also an additional concern in that the energy return on investment (EROI) of most biofuels is very low, typically in the range 0.8 to 1.6, whereas that of petroleum is typically greater than 10 [99,160,161]. The EROI of a fuel is the amount of energy it provides divided by the amount of energy required to produce the fuel. Therefore, it needs to be greater than 1 to provide extra energy to society, but it has been argued that it should be greater than at least 3 for a sustainable society [160].

(2) **Improved public transport**. If more commuters are able to carry out much or all of their transport by sharing public transport systems, then on average this should reduce the total $CO_2$ emissions of the commuters. This is especially so if the public transport in question has relatively low $CO_2$ emissions. Encouraging commuters to cycle or walk instead of driving could also reduce their individual "carbon footprint". Unfortunately, we note that the development of public transport facilities can sometimes conflict with other interests. For example, when Dublin City Council (Ireland) planned to widen the routes on a number of radial routes to provide for bus and cycle lanes, a public outcry was raised out of concern for the mature roadside trees that would need to be felled to facilitate the project [162,163]. While suburban rail networks have the potential to reduce carbon emissions while also saving energy and money, these are susceptible to the risks of cost overruns, project delays and benefit shortfalls that go with large infrastructure projects [164]. Meanwhile, rural communities often cannot be adequately served by these public transport systems. As a result, policies which promote public transport systems over cars can be biased against rural dwellers.

(3) **Use of electric vehicles (EVs)**. If the electricity used to power an EV is produced by wind, nuclear, solar, or hydro power, then the $CO_2$ emissions are significantly lower than those from internal combustion engine vehicles. Therefore, in countries such as Norway, Iceland, and Costa Rica, a motorist who switches to driving an EV could dramatically reduce their personal "carbon footprint" [165]. That said, if the $CO_2$ emissions from electricity production are high, then driving an EV could well increase total $CO_2$ emissions. For instance, Asaithambi et al. (2019) have calculated that EVs used in China produce higher $CO_2$ emissions than a regular car, although for the US, Germany, and Japan, the average emissions for an EV were lower than that of a regular car [166]. On the other hand, Onat et al. (2015) calculated by analyzing the state-wide electricity generation mixes that EVs are the least carbon-intensive vehicle option in only 24 of the 50 United States [167]. Still, we can appreciate why the sale of EVs is being promoted as environmentally desirable. However, we remind readers of the discussion of mineral scarcity in Section 3.3.3.

Often these measures are referred to collectively as "sustainable transport". As can be seen from Figure 1 and Table 1, 10% of the global climate change expenditure over 2011–2018 has been on "sustainable transport" with a further 1% specifically on "biofuels".

## 4.2. Climate Change Caused by Wind Farms

The main rationale for the substantial increase in wind farm installations, in order to reduce the impact of human-caused global warming from greenhouse gas emissions (as described in Section 4.1),

is called into question by the fact that not all climate change is global climate change; there can also be local and regional climate change. Furthermore, there is more to climate change than merely temperature change, and there are other drivers of climate change, beside atmospheric concentrations of greenhouse gases. It is widely acknowledged that land use and land cover changes (LULCC), such as deforestation, and land management changes (LMC), such as irrigation, can affect climate from local to global scale through physical and chemical interactions between land and atmosphere [168]. Thus, there is a risk that any large-scale energy installation that involves changes in land use, land cover or land management can potentially cause local, regional, and global climate change.

In particular, recent years' research has produced considerable theoretical and empirical evidence that wind turbines can have significant local or regional effects on climate. For example, Abbasi et al. (2016) [59] explain that "large-scale wind farms with tall wind turbines can have an influence on the weather, possibly on climate, due to the combined effects of the wind velocity deficit they create, changes in the atmospheric turbulence pattern they cause, and landscape roughness they enhance".

Therefore, before assuming that increasing the deployment of wind farms will "reduce climate change" by only considering the expected reduction in carbon dioxide emissions relative to fossil fuel usage, it is important to compare this expected reduction in "global climate change" to the extra local and regional climate changes it causes. By their very nature, wind turbines have an impact on at least three aspects of the local weather, and thus climate: (1) temperature, (2) wind, and (3) precipitation. In this subsection, we will briefly review what is currently known about the local and regional climate changes caused by wind farms. However, we stress that this is still an emerging subject of research, partially because the increase in wind farm developments in recent years is unprecedented, and the environmental impacts of wind farms have only recently started to garner significant research attention.

Further research is required to extend the evidence base for the impacts of other energy technologies such as large solar power plants [169] and hydroelectric dams [170] on climate through their effects on LULCC and/or LMC.

### 4.2.1. Local Temperature Changes Caused by Wind Farms

Wind farms cause an increase in the average ground and soil temperature downwind from the turbines at night, through a mechanism described schematically in Figure 11, which is adapted from Armstrong et al. (2014) [171]. In essence, increased turbulence causes increased mixing of the upper and lower atmosphere on the lee side of the turbines. This tends to cause a slight cooling at ground level by day and a warming effect at night.

Several studies have attempted to simulate the climate changes caused by wind farms by comparing a simulation with a large hypothetical wind farm to a control simulation without the farm [60,61,64,172,173]. Although Fitch (2015) argued that the globally averaged mean temperature effects would be very small and the annually averaged local warming modest [173], Wang and Prinn (2010) found that "using wind turbines to meet 10% or more of global energy demand in 2100, could cause surface warming exceeding 1 °C over land installations" [61]. Similarly, Miller and Keith (2018) [64] found that, if the US were to meet all of its current electricity usage with wind-generated electricity, it would lead to an average warming of 0.54 °C for the wind farm regions, and 0.24 °C when averaged over the entire US continent. By comparing this to US temperature projections according to IPCC climate models, they argued this would imply that, "if US electricity demand was met with US-based wind power, the wind farm array would need to operate for more than a century before the warming effect over the Continental US caused by [the wind farms] would be smaller than the reduced warming effect from lowering [$CO_2$] emissions" [64].

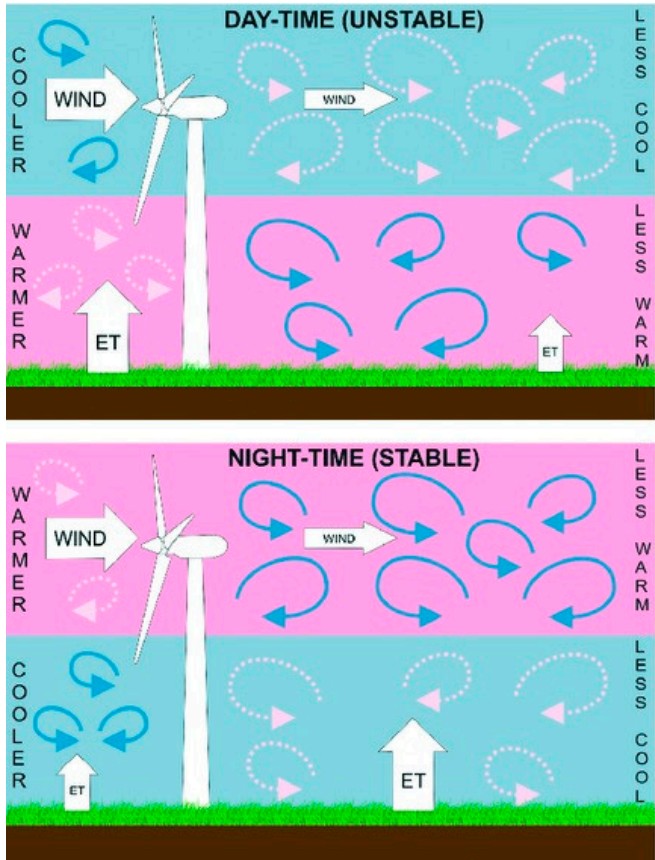

**Figure 11.** "Schematic of the potential effects of wind turbines on air flow, temperature and evapotranspiration during the day with a stable atmospheric boundary layer and at night with an unstable atmospheric boundary layer. The pink (lighter grey) background represents warmer air and blue (darker grey) cooler air. Pink dashed arrows indicate warmer air eddies, which downwind of the turbine are mixed into the cooler air, thus increasing night-time surface temperature. Conversely, the blue solid arrows symbolize cooler air eddies which cause a cooling at the surface during the day-time. The horizontal arrows symbolize the wind speed up and downwind of the turbines, with a reduction in wind speed during the day and night. The vertical arrows suggest hypothesized changes in evapotranspiration, with increases under stable conditions and decreases under unstable conditions downwind of the turbine."—Caption and figure adapted from Figure 1a of Armstrong et al. (2014). Reproduced under Creative Commons copyright license CC BY 3.0; https://creativecommons.org/licenses/by/3.0/.

To complement these modeling-based studies, several studies in recent years have tried to estimate the temperature changes caused by wind farms experimentally, e.g., using field studies and/or satellite-based comparisons. We refer to Abbasi et al. (2016) [59] and Miller and Keith (2018) [64] for summaries of the literature. Figure 12 illustrates one case study, which is adapted from a satellite-based analysis of a region in West Texas (US) by Zhou et al. (2012, 2013) [62,63]. Over an 8-year period, the installation of a large number of wind farms in the region led to a long-term night-time warming of ~0.72 °C/decade in the summer and ~0.46 °C/decade in the winter, for the wind farm regions relative to the surrounding regions. Equivalent studies in Iowa (US) [65] and northwestern China [66] obtained similar results.

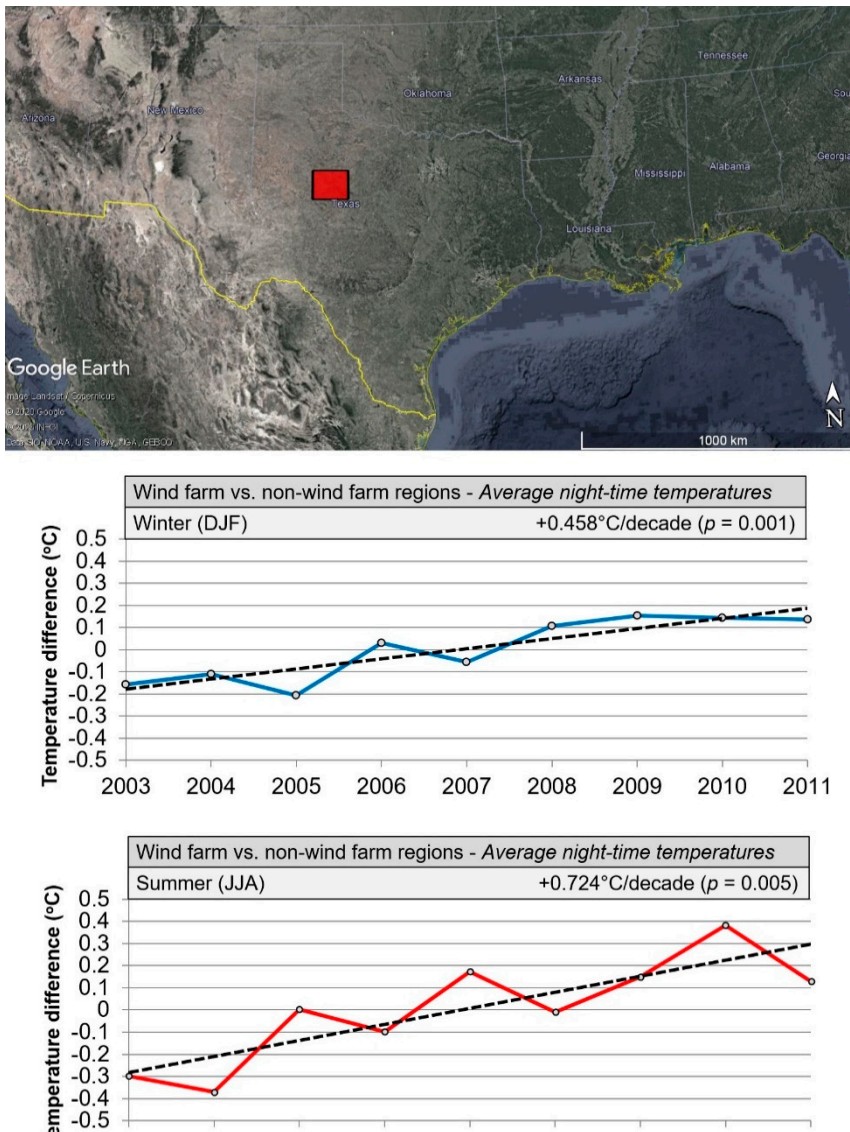

**Figure 12.** Summary of some key results from the Zhou et al. (2012) [62]; (2013) [63] studies on the effects of wind farms on regional land surface temperatures in west-central Texas (US). The top panel shows the approximate location of the study region which was ~10,005 km² (~112.8 km × ~88.7 km) in area. Reproduced in accordance with the attribution guidelines for Google Maps and Google Earth, https://www.google.com/permissions/geoguidelines/attr-guide/, Google, 2020. The middle and bottom panels show the increasing average night-time soil temperatures of the wind farm regions relative to the surrounding regions over the period 2003–2011 for winter and summer respectively. Adapted from Figure 1 of Zhou et al. (2012) [62].

### 4.2.2. Changes in Wind Patterns Induced by Wind Farms

The fact that wind farms influence local wind patterns is, intuitively, the most obvious. The "wake" of a wind turbine, i.e., the wind on the lee side (i.e., downstream) of the turbine, is generally associated with an increase in turbulence and with a decreased wind speed. Although accurate modeling of these effects is surprisingly challenging [174–176], the existence of the "wake effect" is now empirically well established [177–181]. Particularly for off-shore wind farms (which have lower surface roughness than land-based wind farms), this wake effect can cover quite long distances. For instance, Platis et al. (2018) found evidence of significantly reduced wind speeds up to 70 km downwind of a German off-shore

wind farm [179]. This can cause significant economic problems when multiple wind farms are built in the same region, as neighboring farms can end up competing for the same wind [180].

It has also been suggested that these local wake effects might lead to mesoscale changes in weather circulation patterns, especially if wind farms continue to increase in size and number. For instance, Barrie and Kirk-Davidoff (2010) [60] ran a general circulation model simulation in which they simulated what might happen if a hypothetical super-sized wind farm were installed with a 2.48 TW capacity. Their modeled farm would occupy 23% of North America and so was strictly hypothetical. However, this would still only meet 6% of the estimated world electricity usage by 2100 [61], and so is worth considering if the idea of providing a large fraction of the world's electricity from wind farms is to be taken seriously. Their simulations suggest that "the induced perturbations involve substantial changes in the track and development of cyclones over the North Atlantic, and the magnitude of the perturbations rises above the level of forecasting uncertainty" [60]. That is, their hypothetical wind farm could potentially lead to substantial changes in weather circulation patterns. Fiedler and Bukovsky (2011) also found substantial effects in their simulation using a much smaller hypothetical wind farm with a 0.457 TW capacity [172]. They even suggested that with such large wind farms might be capable of altering the paths of hurricanes, but warned that the effects could be somewhat unpredictable without substantial improvements in weather forecasting capability.

### 4.2.3. Local Precipitation Changes Caused by Wind Farms

The effects of wind turbines on local precipitation patterns are less intuitive and have not received as much research attention as yet. However, we can get an intuitive understanding of some of the mechanisms by first considering that the purpose of a wind turbine is to extract mechanical energy from the incoming wind, for conversion into electricity. In other words, the wind downstream contains less energy. Meanwhile, the relative humidity of air is also a function of the energy content of the air, i.e., air temperature. Therefore, by extracting mechanical energy from the incoming wind, the turbines have the potential to alter the relative humidity of the downstream wind. Two visual examples of this can be seen in Figure 13, corresponding to two substantially different atmospheric conditions over the North Sea, off the coast of Denmark [177,178].

We have been unable to identify much research in the literature systematically quantifying the effects of wind farms on local precipitation. However, anecdotally, we have been informed of several incidents of flash-flooding events occurring near wind farms, which were uncharacteristic of precipitation patterns in the area before the construction of the wind farms. From their simulation of the effects of a large-scale 0.457 TW hypothetical wind farm (mentioned above), Fiedler and Bukovsky (2011) [172] noted that, at a local level, "the presence of a wind farm can trigger difference between drought and deluge for the season", but they noted that these effects were less pronounced when averaged over larger regional areas. Nonetheless, they simulated an average 1% increase in precipitation in an area spanning multiple states. Therefore, the effects of wind farms on local and regional precipitation could be quite substantial, and are worthy of more field research.

### 4.2.4. Increase in Biological $CO_2$ Emissions Caused by Wind Farms

Although the warming effects of wind farms described in Section 4.2.1 are mostly localized and tend to be confined to night-time temperatures, we note that they introduce a problematic complication for those proposing to use wind farms to reduce global $CO_2$ emissions. It is true that electricity generation is currently a major component of the anthropogenic $CO_2$ emissions, and therefore reducing the amount of electricity generated using fossil fuels should reduce that component. However, the annual biological $CO_2$ emissions from soil respiration are at least ten times greater than the total annual anthropogenic $CO_2$ emissions [6,182,183].

Typically, the annual emissions from soil respiration are roughly balanced by the absorption of $CO_2$ via photosynthesis through the Net Primary Production (NPP) of the terrestrial plants and trees. However, the total emissions from soil respiration are known to increase with temperature. Estimates of

the exact rates of increase vary between studies, and there are many complexities in extrapolating from the results of e.g., a mid-latitude forest [184] or a tropical region [185] to global estimates (see Davidson and Janssens (2006) for a good review of the challenges involved) [186]. Nonetheless, most studies suggest that the warming of soils generally leads to an increase in biological $CO_2$ emissions from soil respiration [182–187]. Therefore, given that the global $CO_2$ emissions from soil respiration are an order of magnitude greater than anthropogenic emissions, we suggest that the increase in biological $CO_2$ emissions caused by wind farms warming the night-time soil temperatures could potentially be similar in magnitude to the reduction in anthropogenic $CO_2$ emissions from the wind farms.

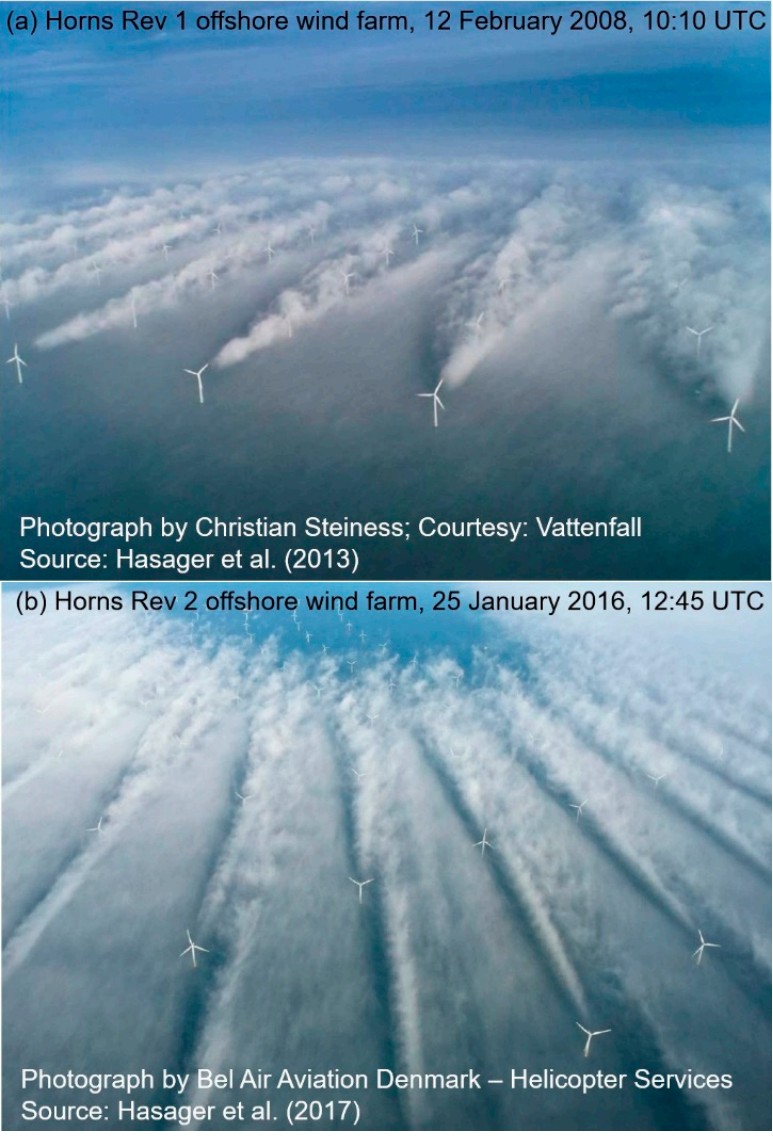

**Figure 13.** Visually striking examples of two different versions of the "wake effect" as observed on separate dates at two neighboring off-shore wind farms off the shores of Denmark, i.e., Horns Rev 1 and 2. (**a**) Photograph by Christian Steiness of an example of a wake effect caused by cold humid air passing over a warmer sea surface, adapted from Figure 2 of Hasager et al. (2013) [177]. Reproduced under Creative Commons copyright license CC BY 3.0; https://creativecommons.org/licenses/by/3.0/. (**b**) Photograph by Bel Air Aviation Denmark—Helicopter Services of an example of a wake effect caused by warm humid air passing over a cooler sea surface, adapted from Figure 2 of Hasager et al. (2017) [178]. Reproduced under Creative Commons copyright license CC BY 4.0; https://creativecommons.org/licenses/by/4.0/.

### 4.3. Reducing Air Pollution

Most of the energy technologies, especially those based on combustion processes, also produce small amounts of undesirable air pollution during their usage. The main forms of air pollution of concern are as follows.

- Particulate matter (PM). This includes larger soot and smoke particles as well as microscopic particles which are often divided into particles less than 10µm in size ($PM_{10}$) and those less than 2.5µm in size ($PM_{2.5}$). The term "black carbon (BC)" is used to refer to PM that is composed of only carbon.
- Various oxides of nitrogen, collectively referred to as $NO_x$
- Sulfur dioxide ($SO_2$)
- Carbon monoxide (CO)—not to be confused with $CO_2$
- Ground-level ozone ($O_3$)—not to be confused with the stratospheric ozone which is found in the "ozone layer"
- Volatile organic compounds (VOCs)

We believe it is important to stress the difference between air pollution and greenhouse gas emissions, because in popular culture they are often mistakenly conflated. As a result, many people assume that policies which aim to reduce greenhouse gas emissions are synonymous with reducing air pollution, and vice versa. Indeed, we note that media articles and reports on climate change and/or greenhouse gas emissions will often include images or video footage of scenes of air pollution (or sometimes footage of steam exhausts coming from industry), see, e.g., in [188]. This may be because media images or footage of carbon dioxide ($CO_2$) are physically impossible as $CO_2$ is an invisible, odorless, tasteless gas. We also stress that due to its role in the photosynthesis/respiration cycle, $CO_2$ is a source of fertilization driving an increase in global greening [72,189], regardless of its relevance as a driver of climate change. Therefore, in this subsection, we are explicitly excluding the greenhouse gases discussed in Section 4.1 from what we consider "air pollution".

A lot of literature exists collectively arguing that all of the forms of air pollution listed above are harmful to human health [190–194]. However, the reliability, reproducibility, and/or statistical robustness of many of the studies that claim to have identified causal links with disease outcomes have been questioned [194–196]. For example, from a review of meta-analyses to elucidate associations between ambient air pollutants and various health outcomes, Sheehan et al. (2016) list 30 meta-analyses indicating only modest if any increases in mortality associated with exposure to the pollutants above [194]. Mindful that multiple hypothesis testing and multiple modeling, p-hacking, and publication bias can result in false-positive effects becoming established fact, Young and Kindzierski (2019) evaluated a highly cited meta-analysis paper examining whether air quality exposure triggers heart attacks. They found that the conclusions of that paper did not withstand critical scrutiny because the shapes of the *p*-value plots were consistent with analysis manipulation in some of the base papers [196]. From time series analysis of a large dataset for air quality and acute deaths in California, Young et al. (2017) found no association between ozone or $PM_{2.5}$ and acute deaths, and therefore no evidence of causal effect in California. They found that daily death variability was mostly explained by season or weather variables [195]. Therefore, we advise the reader to treat cautiously the many claims to have identified causal links between air pollution and human disease outcomes.

Given the fact that all of the above air pollutants are naturally occurring, policy-makers should be wary of so-called "zero tolerance" policies with regards to air pollution. All of the above forms of "air pollution" would be present in the atmosphere to some degree even in the absence of the human species, so measures to attempt to completely eradicate all "air pollution" are physically impossible.

Nonetheless, most people would probably agree that severe pollution is at the very least unpleasant. For this alone, policies that reduce air pollution in regions with air quality concerns can often have considerable public support. We have identified three key sources of air pollution which seem to be particularly relevant for energy policy:

(1)   Air pollution from electricity generation (chiefly from coal-burning plants) and industry
(2)   Air pollution arising from transport, particularly during traffic
(3)   Air pollution from the household burning of solid fuels (chiefly biomass, but also coal)

All three are issues that can cause air quality problems in urban areas, but the third issue is also a major concern for many rural communities in the developing world, due to indoor air pollution.

4.3.1. Air Pollution from Electricity Generation

Electricity generation is often associated with some air pollution from the exhausts of the generating plants, e.g., oxides of nitrogen ($NO_x$), sulfur dioxide ($SO_2$), particulate matter (PM) including soot, volatile organic compounds (VOCs), and ozone. Turconi et al. (2013) [197] carried out a detailed meta-analysis review of 167 studies to estimate the life cycle average emissions of the first two of these ($NO_x$ and $SO_2$) from most of the main forms of electricity generation. We have plotted the results in Figure 14.

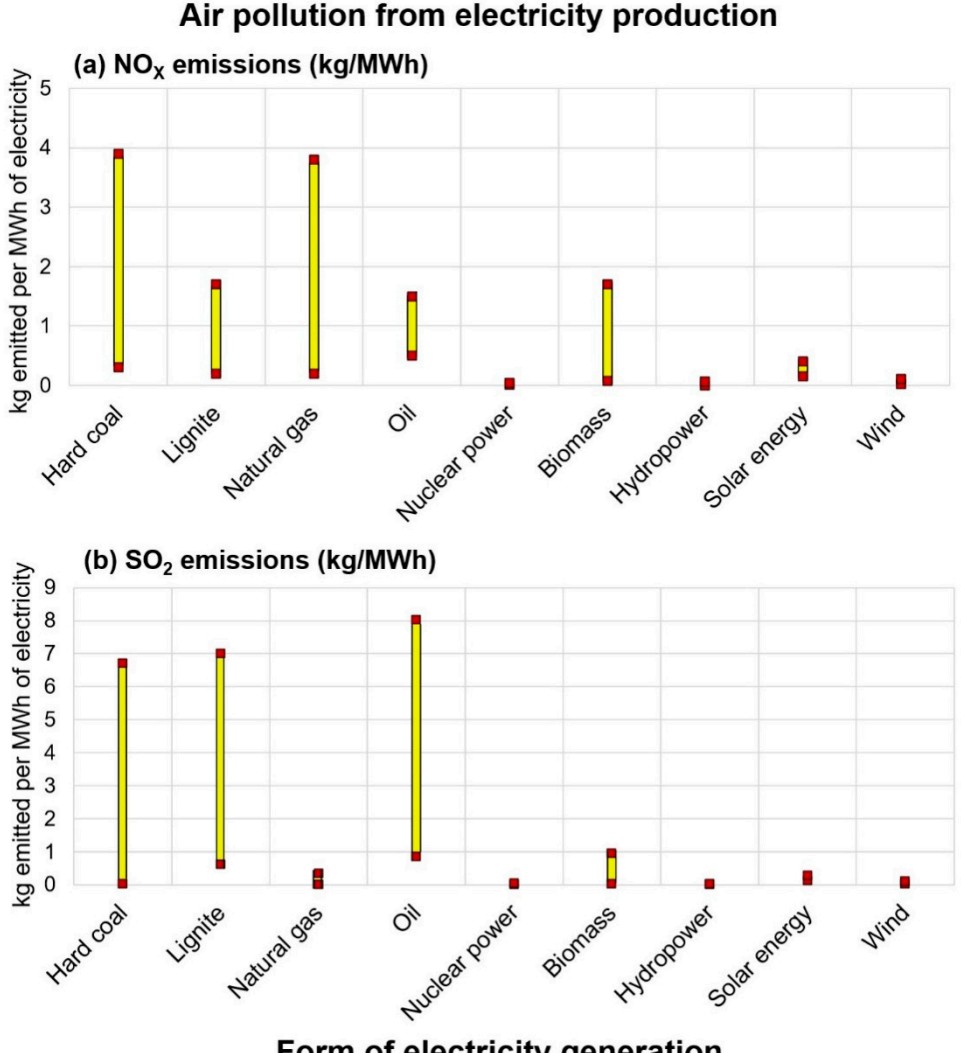

**Figure 14.** Estimates of the (**a**) $NO_x$ and (**b**) $SO_2$ emissions from electricity generation using different types of generation. Data from Table 2 of Turconi et al. (2013) [197].

Turconi et al. (2013) compared the emissions of $NO_x$ and $SO_2$ of electricity generation from fossil fuels, nuclear, and renewable energy. According to their data, coal (whether lignite or hard coal) and oil produce both $NO_x$ and $SO_2$ in considerable quantities, while natural gas and biomass produce considerable quantities of $NO_x$, but not as much $SO_2$. On the other hand, nuclear power, hydropower,

and wind produce very low emissions of both pollutants, and the emissions associated with solar energy are modest.

The use of coal, oil, natural gas (i.e., the fossil fuels), and to a lesser extent biomass (one of the renewables) for electricity production is associated with air pollution from $NO_x$ and $SO_2$. Therefore, one approach to reducing these emissions could be to transition to some combination of nuclear power, hydropower, wind, and to some extent solar power. However, another approach is to reduce the pollutants that are emitted before they leave the plant [134]. Several technologies have been developed to remove air pollutants from emissions from coal-fired thermal power plants, including wet scrubbers, electrostatic precipitators (ESP), and fabric filters to remove particulate matter; selective catalytic reduction (SCR) and selective non-catalytic reduction (SNCR) to remove oxides of nitrogen ($NO_x$); and wet flue gas desulphurization (WFGD) to remove sulfur dioxide [198]. Scrubber systems for use on smaller-scale biomass combustion and gasification systems have also been developed [199,200].

Therefore, air pollution can be largely avoided, even when using fossil fuels or biomass, provided that such scrubber-type systems are installed and operated. However, these systems do increase the cost of electricity generation, and so they are still not widely implemented in developing nations that are prioritizing minimizing cost over minimizing pollution. With that in mind, an alternative method to reducing (but not eradicating) air pollution could be to transition from coal to natural gas. As can be seen from Figure 14, this substantially reduces $SO_2$ emissions and can sometimes partially reduce $NO_x$ emissions [131,132]. It has also been shown to substantially reduce emissions of PM, smoke and smog [133]. Another approach to reducing $SO_2$ emissions is to use low-sulfur content coal instead of the more common (and typically cheaper) high-sulfur content coal [201].

### 4.3.2. Urban Air Pollution from Transport

One of the main sources of air pollution in urban areas is that arising from the transportation sector, e.g., cars, trucks, as well as public transport [202–206]. We note that pollution from traffic has probably always been an issue for urban areas, although its form can change. For instance, in the 19th century, the build-up of horse manure from horse-powered transport was a growing concern, particularly for busy cities [207]. Therefore, many of the main policies for reducing urban air pollution focus on the transport sector.

In industrialized countries, regulatory controls on vehicle exhaust fumes have successfully reduced emissions of nitrogen oxides, carbon monoxide, volatile organic compounds and particulate matter [204]. However, in many developing countries, air pollutant emissions have been growing strongly (Uherek et al., 2010) [202]. Furthermore, even in developed countries with regulatory controls on emissions, the large quantity of vehicles in built-up areas (particularly those with traffic problems) can reduce the air quality.

One approach is to encourage commuters and urban dwellers to walk or cycle more often instead of driving. This may also have added health co-benefits by encouraging people who might otherwise have relatively sedentary lifestyles to exercise more. A related approach is to encourage more use of public transport. As discussed in Section 4.1, if large numbers of commuters are able to carry out much of their travel via shared public transport (e.g., buses, trams, or trains), then this may reduce the total hourly emissions for the area. That said, it is worth remembering that if, for example, too many roads or road lanes are set aside for bicycle lanes or bus corridors, this may increase traffic congestion among the remaining motorists, potentially increasing emissions, because exhaust emissions per kilometer travelled can increase when cars or trucks are idling and stopping/starting due to traffic congestion.

With that in mind, there is much focus on reducing the emissions from vehicles through, e.g., changing fuel types. However, changing fuel type can often reduce one form of air pollution only to increase others [203–206].

Another approach might be to encourage motorists to switch to using electric vehicles or hybrid vehicles [165–167]. However, again, we remind readers of the discussion in Section 3.3.3, and particularly the observations of Herrington et al. (2019) [58] and Mills (2020) [45] that the amount

of limited materials such as cobalt and lithium which would be required to switch even a small fraction of the ~2 billion cars to EVs is enormous.

### 4.3.3. Air Pollution from the Household Burning of Solid Fuels

If a large number of houses in an urban area rely on burning solid fuel (e.g., coal) for heating and/or cooking, then this can significantly contribute to urban air pollution, including haze and smog events [193,208,209]. It can also cause substantial indoor air pollution, and this is a major problem for many rural communities, especially in the developing world. Approximately 3 billion people in the world rely on solid fuels most of their household energy needs, of which ~2.4 billion use biomass (mostly wood, charcoal, animal dung, or crop wastes), while the rest use coal [190,191,210–213]. The majority of these people are in rural communities in developing nations.

The reliance of these households on biomass for most of their energy needs means that nominally they may appear to be major "renewable energy" advocates. However, mostly the reality is that it is not an intentional choice. Approximately 1.3 billion people still do not have a connection to electricity [213], and for many in poverty, the use of biomass as a fuel for cooking, heating, and/or lighting is a pragmatic necessity [210]. For instance, Gupta (2020) notes that, "In Ethiopia, more than 95% of the households depend on biomass energy for cooking and over 70% do not have access to reliable electrical energy at least for basic purposes (lighting and appliances)" [210]. This use of wood and charcoal as a primary fuel is a significant contributor to tropical deforestation, especially on the African continent [214,215]. This reliance on the burning of solid fuel indoors also means that many rural householders are exposed to considerable amounts of indoor air pollution on a daily basis [190,191,210–213].

One way to reduce the amount of this indoor air pollution would be to help householders with low-quality cookers and/or fuels to switch to improved products which generate less pollution [210,212,213]. However, broadly, the main problem is one of poverty and/or lack of access to electricity. It has been suggested that there is a ladder of the main household energy source with increasing income, roughly as follows, crop residue/animal dung → wood → charcoal → kerosene → liquefied petroleum gas (LPG) → electricity [213]. Therefore, we suggest that the most straightforward policies for reducing the worldwide problem of indoor air pollution would be those that help developing nations out of poverty and/or provide electricity supplies to those who do not yet have one.

### 4.4. Protecting Biodiversity

McDonald et al. (2009) [53] caution that, while many studies have quantified the likely impacts of climate-driven habitat loss on biodiversity, relatively few studies have evaluated the habitat impact of spatial extent of energy production, or "energy sprawl". Biodiversity is defined as a contraction of the term: biological diversity", referring to the range of variety among and between living organisms [216]. The places, spaces, or zones where organisms live are known as "habitats" [217]. In general, energy production can have impacts on biodiversity through land use and land cover change (LULCC), air quality, and water quality [53]. LULCC tends to give rise to habitat replacement and habitat fragmentation, which tend to scale with areal impact, while air quality and water quality impacts tend not to.

Energy sprawl is defined as the product of annual energy production (e.g., TWh/yr) and land use intensity of energy production (e.g., km$^2$ per TWh/yr). Since land use intensity is the inverse of power density, energy sprawl is inversely proportional to power density and varies by three orders of magnitude. Thus, the energy sprawl associated with nuclear power and coal is least, while that associated with biomass-generated electricity and biofuels is several hundred times greater, and that associated with wind, hydro and solar power is intermediate [53].

In addition, McDonald et al. (2009) [53] point out that some energy production technologies involve clearance of all natural habitat within their area of impact. This is true for nuclear, coal, solar, hydro power, and growth of biomass or biofuel crops. Other energy production technologies have a relatively small infrastructure footprint, with larger areas impacted by habitat fragmentation. This is true of techniques that involve wells, e.g., geothermal, natural gas, and oil, for which about

5% of the impact area is due to direct land clearance, while 95% is due to habitat fragmentation and species avoidance behaviors. Similarly, ~3–5% of the impact area of wind turbines is due to direct land clearance, while 95% is due to habitat fragmentation, species avoidance behaviors, and bat and bird mortality.

Through an extensive review of the literature regarding the impacts of renewable energies on ecosystems and biodiversity, citing hundreds of earlier authors, Gasparatos et al. (2017) [55] identified the main mechanisms of ecosystem change and biodiversity loss for renewable energy pathways including solar, wind, and bioenergy, along with interventions to mitigate their adverse impacts. These are outlined with regard to bioenergy, hydro, solar, and wind, below.

Direct and indirect land use change from the expansion of biomass feedstock for energy production have resulted in habitat and biodiversity loss, especially when large-scale land conversion using monocultural feedstock production is adopted [55]. In addition, the authors cite several life cycle assessments (LCAs) which have demonstrated that most biomass energy production pathways emit air and water pollutants that can have negative effects on biodiversity via eutrophication, acidification and toxicity. Atmospheric emissions from key biomass energy species such as eucalyptus, poplar and willow contribute to tropospheric ozone formation, which is harmful to plant life.

They cite several studies that offer examples of negative biodiversity outcomes of biofuel-driven habitat loss and change around the world. For example, oil palm cultivation in Southeast Asia has mainly replaced primary/secondary tropical forests rather than agricultural land. In the US, soybean for biodiesel and corn/sugarcane for bioethanol will have a consistently larger effect on future land use change than other renewable energy pathways [55]. Fargione et al. (2010) estimate that biodiversity is reduced by ~60% in US corn and soybean fields, and by ~85% in Southeast Asian palm oil plantations, relative to unconverted habitat [54].

The use of fertilizers/agrochemical runoff and industrial effluents from biofuel production are major sources of water pollution in Brazil and Southeast Asia. Ecotoxicity effects due to pesticide use can also pose risks to biodiversity [55].

The New York Times Magazine's 2018 coverage of palm oil grown to meet a US biofuels mandate illustrates the controversy that can befall the policy-maker advocating in favor of biofuels [218]. The feature is headed: "Palm oil was supposed to help save the planet. Instead it unleashed a catastrophe." While the office of Nancy Pelosi, Speaker of the House of Representatives, has defended the mandate, arguing that biodiesels are cleaner than fossil fuels, Representative Henry Waxman argues that Congress was so focused on domestic climate policy that it failed to see the repercussions of its climate policies around the world. "We're doing more harm to the environment," Waxman says. "It was a mistake" [218].

The environmental impacts of hydro power depend on the scale and type of power installation involved. Large hydro, involving a reservoir created by damming a river, has much greater impact than "run-of-river" (ROR) installations, which may use a small dam to generate a head of water, or installations using in-stream turbine technology, which do not rely on damming the river [155]. River dams constructed for hydro power or other uses have impacts on seasonal variation in downstream flow and on the transport and transformation of nutrients including carbon (C), nitrogen (N), phosphorus (P), and silicon (Si) along the length of the river from the reservoir to the ocean. The interruption of seasonal flows can have adverse impacts on aquatic and estuarine flora and fauna through alterations in water depth, salinity, and temperature. Alterations in nutrient concentrations and ratios can cause eutrophication and harmful algal blooms (HAB's) in coastal zones [219].

Finer and Jenkins (2012) lament a lack of strategic planning with regard to the regional and basin-scale assessment of potential ecological impacts of hydropower [156]. From an ecological impact analysis of 151 dams of greater than 2 MW planned to be constructed over the following 20 years, they classified 47% as high-impact and just 19% as low-impact, and they estimated that 80% would drive deforestation due to new roads, transmission lines or flooding of reservoirs. Sixty percent of

the dams would cause the first ever major break in connectivity between Andean headwaters and the lowland Amazon.

Mitigation measures to reduce the impacts of hydropower installations on ecosystems and biodiversity include (a) selecting hydropower technologies that have less severe impacts, (b) using biodiversity-friendly elements such as bypass flows, and (c) implementing innovative policies such as regulatory measures [55]. Moran et al. (2018) indicate that the environmental impacts of hydro dams can be mitigated through hydropower installations that utilize in-stream turbine technology, also known as "zero-head", sited in rivers with flow velocity exceeding 1 m/s to produce steady baseload power and avoid intermittency associated with variation in seasonal flows [155]. For best outcomes, they recommend that environmental impact assessments (EIA) and social impact assessments (SIA) be conducted by independent firms rather than dam construction companies. Almeida et al. (2019) point out that steeper topography favors higher power densities [220]. Dams in mountainous areas of Bolivia, Ecuador, and Peru tend to have higher power densities than lowland Amazon dams of Brazil.

Utility-scale solar energy (USSE) can affect ecosystems in multiple ways throughout its life cycle (i.e., construction, operation, and decommissioning) [55]. Habitat loss can arise from solar power infrastructure, particularly USSE, as it occupies substantial tracts of land. Supporting infrastructure (e.g., access roads and electrical equipment) and spacing between panels can result in actual space requirements being approximately 2.5 times the total area of the panels themselves. Solar energy installations have also been associated with pollution of land and water because cleared land is often maintained with dust suppressants and herbicides [55].

Proposed mitigation measures to reduce the impacts from the deployment of solar energy on ecosystems and biodiversity include (a) locating USSE installations in areas with little biodiversity and (b) developing biodiversity-friendly operational procedures. USSE installations can sometimes be developed in desert areas that combine high levels of solar insolation with relatively little cloud cover and biodiversity. However, some desert ecosystems host highly specialized and rare species that are known to be particularly vulnerable to human activity [55]. Some of the impacts on habitats can be reduced by mounting solar PVs on rooftops and building facades, e.g., in urban settings, because solar panels mounted on existing structures do not convert or fragment habitats [55].

Wind energy installations can have a number of ecological impacts on avian and aquatic species, depending largely on whether they are located on shore or off shore [55]. Despite many improvements in wind turbine design, wildlife mortality, especially among birds of prey, remains high [221,222]. On-shore habitat loss may arise through bird and bat species avoiding areas containing wind generators. Habitat change can arise from the collision of birds (particularly raptors) and bats with the wind generators [221–224]. An estimated 234,000 birds are killed annually from wind turbines in the United States alone [222]. Bats suffer more than birds, with the impact estimated to be to the order of tens of bat fatalities per turbine per year. The construction of offshore turbines is hazardous to marine mammals, especially because of noise generated during pile-driving which can be heard at distances of up to 80 km under water [225,226]. Marine mammals often avoid areas of underwater construction, only slowly returning after construction is completed [225]. On a more positive note, once established, the foundations of the turbines can be colonized by marine life creating an artificial reef or sanctuary [227].

Common mitigation measures to reduce the impact of wind energy generation on ecosystems and biodiversity include (a) locating wind power installations in areas of little biodiversity and (b) developing biodiversity-friendly operational procedures for wind energy generation. In contrast to solar power, the most suitable places to locate wind turbines may also be places that could cause most damage to avian biodiversity. For example, while most proposed sites for on-shore wind farms in the UK are located in upland areas, these remote windy locations are also areas of high conservation importance for birds. Biodiversity-friendly operational procedures include minimizing the overall development footprint, e.g., by installing transmission cables underground, and minimizing the risk

of collision, e.g., by making blades more visible or grouping them in configurations aligned with the main flight pathways [55].

On assessing the current and likely future extent of RE production infrastructure associated with onshore wind, solar, and hydropower within conservation areas, Rehbein et al. (2020) identified 2206 fully operational RE facilities located within protected areas, key biodiversity areas, and Earth's remaining wilderness, with another 922 facilities under development [228]. However, Sonter et al. (2020) caution that the impacts of mining activities associated with RE production infrastructure may be more extensive than those of their spatial footprint or other environmental risks. On mapping the global extent of areas potentially impacted by mining, they find that habitat losses associated with future mining for RE could present threats to biodiversity that surpass those averted by climate change mitigation [229].

The fossil fuels and nuclear power likewise have various biodiversity impacts. Acar and Dincer (2019) ranked a range of power sources according to a number of environmental impacts, including biodiversity [230]. They found that coal has a high impact, via its impacts on air quality, water quality and land contamination, while gas has low impact, and nuclear has moderate to high impact on biodiversity. Brook and Bradshaw (2015) [51] concur that, because of its very high power density and small land requirements, nuclear power offers good prospects for baseload power with modest biodiversity impacts.

## 5. Socioeconomic Concerns Associated with the Various Energy Technologies

In Section 4.3.3, we noted that, in the developing world, ~1.3 billion people still do not have access to an electricity supply. Moreover, we noted that ~3 billion people rely on the household burning of solid fuel for most of their energy needs (cooking, heating, and lighting), and that for most of these (~2.4 billion), this fuel usually consists of wood, charcoal, animal dung, or crop wastes [190,191,210–213]. Technically, these "biomass" fuels are "renewable energies", but as discussed in Section 3.3.1, this does not imply that their use is "sustainable". Technically, biomass is considered "carbon-neutral", and therefore the promotion of the use of biomass (and the related "biofuels") is one of the strategies for reducing greenhouse gas emissions (Section 4.1). Indeed, from Table 1, we see that at least 3% of the US$3.66 trillion of global climate change expenditure over the 2011–2018 period has been spent on "biomass and waste" and "biofuels" projects.

Therefore, nominally, it could be argued that, in terms of keeping $CO_2$ emissions low, these developing nations are among the most successful. As discussed in Section 4.1, many currently consider reducing $CO_2$ emissions to be one of the top priorities for the world, especially in terms of preserving the environment. However, the reality is that this apparent "success" has nothing to do with policies to protect the environment, but is chiefly a result of poverty, especially in rural communities. Indeed, the use of biomass as solid fuel in rural communities has been shown to be a significant driver of tropical deforestation, especially on the African continent [214,215].

More broadly, a substantial body of literature has found empirical evidence that the so-called "environmental Kuznets curve" (EKC) appears to apply to many environmental indicators, although not all [136,231–234]. The EKC hypothesis developed in the 1990s partly out of earlier debates between the neo-Malthusians and cornucopians in the 1970s (Section 3.3.1). In 1955, Simon Kuznets proposed an "inverted U curve" relationship between income inequality and economic growth, i.e., that as a country developed economically, income inequality would initially increase, but after some turning point, further economic growth would begin to reduce income inequality again. This became known as the Kuznets curve.

Starting in the 1990s, numerous studies found considerable empirical evidence that for many environmental indicators, particularly those associated with local air pollution (Section 4.3); there appears to be a similar relationship between economic growth and environmental impacts [136,231–234], i.e., the environmental Kuznets curve (EKC). This implies that, in the short-term, encouraging developing nations to develop may lead to environmental degradation, but that, in the long-term, once they have passed

the relevant "turning points", further development will reduce environmental degradation. However, the same analyses which reveal that the EKC applies to local forms of pollution also show that it does not apply to issues that are more global in nature, e.g., $CO_2$ emissions [10,231,232,234].

On the contrary, on average, $CO_2$ emissions appear to increase with economic development. This has led those prioritizing reducing global $CO_2$ emissions to explicitly warn that we should not rely on the EKC to automatically lead to $CO_2$ emissions reductions. Instead, they argue that new paths for development need to be designed which explicitly incorporate $CO_2$ reduction as an additional top priority [10,232,234].

We want to emphasize some important corollaries of the above:

(1)  The goal of reducing global $CO_2$ emissions is directly opposed to the standard pathways of economic development which have been followed historically.

(2)  We stress that this does not in itself preclude the possibility that alternative pathways to economic development which also reduce global $CO_2$ emissions could exist. Indeed, as discussed in Section 4.1, France and Sweden are two notable examples of developed nations that combined economic growth with relatively low $CO_2$ emissions through investment in nuclear [52]. Therefore, research into exploring the possibilities of new pathways to economic development is justifiable and laudable [10,52,232,234]. However, we should acknowledge that new pathways by their very nature will not have been tested to the extent of the standard historic pathways.

(3)  Aside from $CO_2$ emissions, and despite the neo-Malthusian predictions discussed in Section 3.3.1, the EKC studies confirm that the standard pathways to economic development actually lead to reductions in environmental degradation for many aspects of the environment, especially those associated with local pollution.

In other words, the most straightforward routes for helping nations develop and/or reducing world poverty fundamentally conflict with the goal of reducing $CO_2$ emissions. We suggest that even within developed nations, policies to reduce $CO_2$ emissions similarly are often at odds with improving the livelihoods of the less affluent in society.

For example, one policy tool which is often promoted as being potentially useful for reducing $CO_2$ emissions is the implementation of "carbon taxes". Carbon taxes can take many forms, but typically penalize the use of forms of energy that are associated with relatively high $CO_2$ emissions. Researchers studying the socioeconomic implications of various carbon taxes in multiple countries have found that carbon taxes "tend to be regressive", i.e., the burden tends to be greatest on the poorest households [235–240]. That is, while the absolute tax paid by richer households is often greater, as a percentage of their income it tends to be much lower. Suggestions have been made about how to partially mitigate against this regressive nature by, e.g., explicitly coupling the carbon tax with additional tax breaks for lower income groups for other taxes, or redistributing some of the tax revenue directly to lower-income groups via social welfare supplements [235–240]. However, it indicates that carbon taxes have an underlying tendency towards greater income inequality.

Carbon taxes also may be biased against rural dwellers [239–242], e.g., if the carbon tax is designed to encourage the use of public transport systems which do not adequately service rural communities. Indeed, the "Mouvement des Gilets Jaunes" ("Yellow Vest Movement") protest movement in France which began in late 2018 appears to have been motivated by resentment over increasing carbon taxes on motor fuel, which were perceived to be unfairly biased against rural communities that were more reliant on motor transport [241,242]. (The name refers to the yellow high-visibility jackets that car-owners are obliged to keep in their car under recent regulations, and hence were worn as a symbol of the movement.) Prud'homme (2019) notes the irony that France happens to already be one of the most decarbonized developed nations, since the French electricity grid is 85% nuclear and hydroelectric [242].

Chancel and Piketty (2015) note that there is an additional regressive nature to carbon taxes when considered on an international basis [243]. That is, the introduction of the same carbon tax to multiple

countries will tend to create greater burdens on lower income countries. With that in mind, they have proposed the possibility of creating a global "carbon tax" towards a "climate adaptation fund" where the taxes would be greater for higher-emissions countries, and the funds would be mostly distributed to developing nations [243].

There is an additional irony in this conflict between the standard pathways to economic development and reducing $CO_2$ emissions in that developing nations are often the least well-adapted to dealing with climate change and/or weather extremes. For instance, while hurricanes can cause considerable damage when they make landfall in the United States [244], many neighboring nations in the Caribbean or along the Gulf of Mexico are particularly vulnerable [245,246]. Although recent research has confirmed that there has been no long-term trend in the number or intensity of hurricanes making landfall in the area [244,247], the destructive nature of these extreme weather events coupled with the infrequency with which they strike any given region can cause devastating effects. Therefore, investment into "climate adaptation" infrastructure, e.g., improved resilience for hurricanes [245] along with better hurricane response systems can be worthwhile investments in at-risk hurricane zones [67,248]. However, these often require substantial economic investment which can be out of reach for lower-income countries. With that in mind, it is surprising that only 5% of the global climate change expenditure over the 2011–2018 period has been spent on "climate adaptation" projects (Figure 1 and Table 1).

We agree with Pielke Jr. [67,248] and Chancel and Piketty (2015) [243] that greater investment in "climate adaptation" makes sense if society wants to better respond to climate change and extreme weather. However, we also note from the discussion above that one of the key ways to help developing nations to improve their resilience to weather extremes is to encourage their economic development. In particular, having continuous access to an affordable and reliable electricity and energy infrastructure is essential. With that in mind, Epstein (2014) has made the "moral case for fossil fuels" [8], arguing that the standard pathways to economic development making extensive use of coal, oil, and/or gas have been well tested and should be encouraged. Others caution that this would lead to substantial increases in $CO_2$ emissions, and favor the development of nuclear instead [20,21,23,42–44,51,52]. Helm (2018) argues that a temporary transition from coal and oil to gas for a few decades could offer a compromise between the two approaches that would allow time for a slower long-term energy transition [41].

Finally, we note that there are often societal conflicts associated with energy policies when they impact on indigenous peoples without adequate consultation. Klein (2015) describes the struggles of indigenous peoples in Canada and Australia to restrain the fossil fuel industry from degrading their lands and waters [17], but the materials needed for other energy sources also pose a threat of severe adverse impacts on indigenous peoples, such as

- silver mining on the Xinca indigenous peoples of Guatemala [117]
- lithium mining on Atacama communities in Argentina [249]
- cobalt mining on indigenous peoples in Katanga, Democratic Republic of Congo [250]
- uranium mining on the Mirarr people of Australia's Northern Territory [251].

Hydroelectric dams can likewise have severe impacts on the Munduruku [252] and other indigenous peoples throughout the Amazon Basin [156].

Additionally, Gilio-Whitaker (2019) [253] and Estes (2019) [254] have detailed the impacts on Native American land rights of a range of energy industries. Gilio-Whitaker frames the contamination of Indian lands and waters for uranium mining and fossil fuel extraction, along with the flooding of ancestral lands to construct hydropower dams, as processes in the displacement and colonization of Native Americans. Estes [254] similarly documents the history of construction of hydroelectric dams as a driver of dispossession of Lakota people and of coerced population shifts from their traditional lands to urban centers. Both authors have detailed the series of events by which the Dakota Access Pipeline was laid through Native American lands in North Dakota, without the consent of the Standing Rock Sioux Tribe, whose lands and waters are placed at risk of contamination by pipeline leaks. From the perspective of these indigenous scholars, it seems that the settler state consistently engages in

coercive practices to impinge on indigenous lands, regardless of which energy technology is under development. Kelly (2016) notes failure to consult as one of the causes of failure of ambitious projects, and this seems relevant in the context: regardless of which energy technologies we choose, consultation with indigenous peoples is required to safeguard land rights, social equity, and wellbeing [20].

## 6. Discussion

In the introduction, we argued that none of the main energy sources currently available or currently used (Section 2) should be considered as a "panacea". Instead, each energy source has its pros and cons and we recommend that energy policy-makers consider both. In Table 2, we summarize the key engineering and environmental concerns which we considered in Sections 3 and 4, respectively, for each of the main energy sources. For brevity, we have not included the socioeconomic concerns which were discussed in Section 5, but we recommend these also be explicitly considered.

In Section 3.1, we noted that the three "intermittent" (or "non-dispatchable") energy sources, i.e., wind, solar, and tidal, are very unsuitable for societies that require a continuous, on-demand, electricity supply. This is indeed what has been the norm since the age of electrification began in the early 20th century. We urge policy-makers to recognize that policies which rely on any of these three sources as part of their grid will face increasing problems of grid instability with increasing penetration of the network. Although advocates of these three sources imply that these problems can be partially overcome through the use of energy storage technologies and/or major continental-scale transmission networks, this appears to be based more on wishful thinking than pragmatism.

We note that wind farms also cause considerable local climate change (Section 4.2) and can cause problems for biodiversity (Section 4.4). Although wind farms are associated with relatively low direct $CO_2$ emissions (Section 4.1), we suggest that the local night-time soil heating effect of wind farms may be leading to an increase in biological $CO_2$ emissions, which may cancel some (or perhaps all) of the savings relative to other energy sources (Section 4.2.4).

In terms of power density, the three main fossil fuels (coal, oil and gas) and nuclear are orders of magnitude better than any of the renewables (Section 3.2). Currently, those four technologies account for 89% of the world's energy usage (Section 2), so policies which significantly reduce that percentage may potentially lead to engineering problems due to the reduction in power density. We note that the power density of biomass and biofuels is by far the lowest. As a result, policies which significantly increase the use of biomass and/or biofuels will require particularly large land areas. In Section 4.4, we note that this can lead to increased deforestation and major biodiversity impacts.

In Section 4.4, we also note that hydroelectricity can lead to threats to biodiversity as well as contribute to deforestation. There can also be socioeconomic concerns associated with hydroelectricity, due to the displacement of people in the area. In Section 5, we noted that this is a particular concern for indigenous peoples in certain regions, such as the Amazon River Basin.

One of the main limitations of hydroelectricity and of geothermal is that both technologies are heavily dependent on local geography requirements (Section 4.1). Geothermal can be very effective in regions with thermal springs (e.g., Iceland), and hydroelectricity can be very effective in certain mountainous regions with large local rivers (e.g., Norway). However, suitable sites are quite limited geographically.

The three main fossil fuels (coal, oil, and gas) have collectively powered most of the Industrial Revolution since the 19th century, and, as of 2018, they still provide 85% of the world's energy. Because these are finite resources, there is concern about how long society can continue to rely on them. However, as discussed in Section 3.3.2, the estimated known reserves of coal, oil, and gas should provide enough energy at current rates for several more decades at least, and historically the known reserves have continued to expand over time to surprise commentators that have predicted "peak oil", "peak gas", or "peak coal". Therefore, while we should recognize them as finite resources, they are still in plentiful supply—for now, at least.

**Table 2.** Summary of the engineering and environmental concerns associated with each of the main energy sources which were discussed in this review. This table is not meant to be comprehensive or definitive, but merely to provide a quick overview of the main topics described in more detail in the text. For brevity, the socioeconomic concerns discussed in the review were not included.

| | Engineering Concerns | | | Environmental Concerns | | | |
|---|---|---|---|---|---|---|---|
| | Intermittency | Power Density | Resource Depletion | Greenhouse Gas (GHG) Emissions | Air Pollution | Biodiversity Concerns | Other Environmental Concerns |
| *Electricity* | | | | | | | |
| Coal | No | High | Finite, but substantial reserves | High | High | Moderate | Mining impacts |
| Peat | No | High | Finite | High | High | High | Unique biomes |
| Oil | No | High | See "Peak oil" debate | High | Moderate | Low | Possibility of oil spills |
| Natural gas | No | High | See "Peak gas" debate | Moderate | Low | Low | Potential impacts from "fracking" |
| Nuclear | No | High | Finite, but substantial reserves | Low | Low | Low | Waste disposal, possible meltdowns |
| Hydroelectricity | No | Low | Limited by geography | Low | Low | Moderate | Alters local environment |
| Biomass | No | Very low | Requires large land areas | "Carbon-neutral" | High | Very high | Competition with agriculture |
| Geothermal | No | Low | Limited by geography | Low | Low | Low | |
| Solar | Yes | Low | Resource-heavy construction | Low | Low | Moderate | Disposal of waste |
| Wind | Yes | Very low | Resource-heavy construction | Low | Low | Moderate | Causes local climate change |
| Tidal | Yes | Low | Resource-heavy construction | Low | Low | Low | |
| *Transport* | | | | | | | |
| Oil | No | High | See "Peak oil" debate | High | High | Low | Possibility of oil spills |
| Biofuels | No | Very low | Requires large land areas | "Carbon-neutral" | Moderate | Very high | Competition with agriculture |
| EVs | * | * | Major concern | Very low | Very low | Low | Disposal of waste |

\* The relevance of intermittency and power density to EVs depends on the source of electricity.

On the other hand, in Section 4.1, we saw that these fossil fuels are the highest net emitters of $CO_2$ per kWh of electricity, and in Section 4.3, we noted that their use is associated with air pollution, although various approaches have been proposed to reduce the amount of air pollution.

As an aside, we do not include peat among the three fossil fuels mentioned above, as peat resources are relatively limited and only comprise a significant fraction of energy usage in a few places, e.g., Ireland [130], although De Decker (2011) has noted that peat was an important fuel in the pre-industrial Middle Ages for The Netherlands [255].

Finally, nuclear energy has created a lot of public concern, chiefly about potential accidents and/or the safe disposal and management of waste. That said, in Section 4.1, we noted that, while nuclear accidents have on average been the most expensive, they were responsible for only 2.3% of the deaths in energy-related accidents. Moreover, supporters of nuclear argue that the disposal and management of waste can be, and is, satisfactorily resolved.

## 7. Conclusions

Given that all of the energy sources have their advantages and disadvantages, the reader may be wondering which ones to use. We suggest that policy-makers who are trying to decide between various energy policies should consider what their main priorities are, and which priorities they are prepared to compromise on. This may be different for different countries, and may change over time.

For example, suppose a government considers reducing $CO_2$ emissions one of its top priorities. In Section 4.1, we suggested seven different approaches for this, but noted that each conflicts with other priorities (also summarized in Section 6). If protecting biodiversity is also a top priority, then the use of biomass should be avoided, and that of hydroelectricity or wind energy should be treated warily. Meanwhile, if having a stable and reliable electricity supply is also a top priority, then the use of any of the intermittent sources (wind, solar, or tidal) should be minimized, and governments may want to prioritize the use of nuclear, or transition from coal or oil to gas, or invest in carbon capture and storage (CCS) technology.

On the other hand, suppose a government is trying to increase economic growth and/or to improve social equity. In that case, cheap, affordable, and reliable electricity is probably a top priority. Therefore, some combination of coal, oil, gas, and nuclear would probably make sense. If geothermal or hydroelectricity are suitable for the area, they may also be worth considering. If reducing $CO_2$ emissions is also a top priority, then they may want to reduce the amount of fossil fuels they use and develop more nuclear (as France and Sweden have done, for instance), whereas if avoiding the use of nuclear is a higher priority, then they might want to consider using more fossil fuels instead.

Looking at the breakdown in the US$3.66 trillion which has been spent on global climate change expenditure over the period 2011–2018, as described in Figure 1 and Table 1, we saw that 55% was allocated to solar and wind energy projects. This is a very large allocation for two energy sources which have many disadvantages, as summarized in Section 6. Meanwhile, only 5% has been spent on climate adaptation, even though investing in climate adaptation can dramatically improve the ability of societies to deal with climate change and weather extremes. This suggests that global climate change expenditure is not being allocated using a critical assessment of the pros and cons of the key policies. We hope that the analysis in this review can remedy this in time.

**Author Contributions:** All authors contributed to the conceptualization, writing—original draft preparation, review and editing of this paper. All authors have read and agreed to the published version of the manuscript.

**Funding:** C.Ó., G.Q., and M.C. received no external funding for works on this paper. R.C. and W.S. received financial support from the Center for Environmental Research and Earth Sciences (CERES), while carrying out the research for this paper. The aim of CERES is to promote open-minded and independent scientific inquiry. For this reason, donors to CERES are strictly required not to attempt to influence either the research directions or the findings of CERES. Readers interested in supporting CERES can find details at https://ceres-science.com/.

**Acknowledgments:** We would like to compliment the Climate Policy Initiative for their efforts in compiling its annual Global Landscape of Climate Finance reports whose results we used for generating Figure 1 and Table 1, and for providing easy access to these reports on the https://www.climatepolicyinitiative.org/website.

**Conflicts of Interest:** The authors declare no conflict of interest.

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
