# Peer review of "Energy and Climate Policy—An Evaluation of Global Climate Change Expenditure 2011–2018"

_energies, doi:10.3390/en13184839_

Round 1

Reviewer 1 Report

It is worth noting that the topic discussed by the authors is up-to-date and important in the context of the current discussions. The overview of various approaches to energy policy, presented in the introduction, is worth highlighting.

1) Authors should ensure the high quality of the presented figures, i.e. Fig. 1 etc.

2) When choosing a report created by BP, even despite the justification of its comprehensive and detailed nature, one should mention the risks that may result from the analysis of the report published by one of the private corporations. This fact may affect the conclusions. It should be mentioned in the text and the possible risks should be pointed.

3) Figure 7 - please indicate a year

4) Row 262 - different font, please correct

5) The article would benefit from a significant shortening. First of all, in the Introduction part it is worth emphasizing the scientific goals set by the authors, research problem as well as scientific gap identified by the authors. Then the research questions should be listed so that they are clear to the reader. After the Introduction part, you can go to the description and analysis of individual energy sources, but with emphasis on the synthesis of the information provided. Thus, the article will become more readable and transparent. The table in the Discussion section is of great value. However, it would be beneficial to shorten some of the conclusions resulting from the entire article in the discussion, because this part should contain other points of view and discussion with them by the authors of the article.

Author Response

Response to reviewer 1

We would like to thank you for your very helpful and constructive feedback. We have taken your comments on board, along with those of Reviewer 2, and revised the manuscript accordingly. We believe that the revised manuscript has been significantly improved. We hope that you agree, and we thank you for pushing us to make our manuscript stronger and clearer.

Below are our point-by-point responses to your specific suggestions.

1) Authors should ensure the high quality of the presented figures, i.e. Fig. 1 etc.

To ensure the high quality of the presented figures, we have made substantial improvements, as follows. We have increased overall figure size and/or font sizes throughout, to ensure that all captions are readily legible. We have simplified Figure 1 to enlarge the pie chart, while omitting detail that can be found in Table 1. In Figure 6, we have increased line thickness to make trends more vivid, and we have labelled each month on the x axis. In Figure 10, we have expanded the abbreviation “Nat Gas” to read “Natural Gas”. In Figure 12, we have increased the size of the lower panel to make the graphs easier to read. We have increased resolution of all graphic files to comply with the journal’s specifications.

2) When choosing a report created by BP, even despite the justification of its comprehensive and detailed nature, one should mention the risks that may result from the analysis of the report published by one of the private corporations. This fact may affect the conclusions. It should be mentioned in the text and the possible risks should be pointed.

This is a very good point. We have inserted an explicit acknowledgement of the risks that could result from the analysis of a report published by a private corporation that has investments in a range of energy sources. We have also presented figures for global energy mix from a US government report for purposes of comparison and validation to a degree.

3) Figure 7 - please indicate a year

In the caption to Figure 7, we have now inserted the period of years during which the data for mean monthly solar irradiation were gathered, as requested.

4) Row 262 - different font, please correct

Thank you for spotting this. We have corrected the font in row 262.

5) The article would benefit from a significant shortening

We appreciate what you are saying here. We also were aware that our review is quite a long manuscript. However, we suspect that many readers might not share your breath of knowledge of the many interlocking aspects of this challenging subject.

Reviewer 2 also agreed with you that the manuscript is quite long, but necessarily so. Specifically, he/she says, “Some could argue that the paper is too long, but being a review on such a huge domain, I consider the page count to be proper.”.

First of all, in the Introduction part it is worth emphasizing the scientific goals set by the authors, research problem as well as scientific gap identified by the authors. Then the research questions should be listed so that they are clear to the reader. After the Introduction part, you can go to the description and analysis of individual energy sources, but with emphasis on the synthesis of the information provided. Thus, the article will become more readable and transparent.

Although we understand that we could take the approach of the science writer, first listing research questions as a framework on which to present a synthesis of information, thus condensing the contents, we have opted instead to provide a more expansive discussion of each aspect of the energy systems described, to make the paper more readable for policymakers who may not be familiar with all of the disciplines from which we cite.

The table in the Discussion section is of great value. However, it would be beneficial to shorten some of the conclusions resulting from the entire article in the discussion, because this part should contain other points of view and discussion with them by the authors of the article.

We have taken a discursive approach throughout the paper, often presenting opposing points of view. For example, we cite advocates for intermittent energies alongside their critics; we outline neo-Malthusianism alongside Cornucopianism; we cite authors who have expressed concern about peak oil, peak gas and peak oil, followed by others who express reservations about the issue. We consider a broad range of options towards decarbonization and discuss advantages and disadvantages of each one. We discuss the environmental and socioeconomic impacts of fossil fuels and nuclear power alongside those of renewable energies. This distributes the presentation and discussion of opposing points of view throughout the paper, thus alleviating the burden on the Discussion section, which might otherwise need to be very lengthy.

Reviewer 2 acknowledges that the paper presents “pro and against views of different entities and researchers,” concluding: “I found the paper very interesting as it offers in one place the possibility to learn about different dimensions of climate change and energy policy.”

In addition to the improvements listed above, we have conducted a thorough copy-editing review throughout, to ensure consistency of style and compliance with the specifications of the journal.

Author Response

Response to reviewer 2

We would like to thank you for your very helpful and constructive feedback. We have taken your comments on board, along with those of Reviewer 1, and revised the manuscript accordingly.

In particular, you recommended that the quality of the figures should be improved. Reviewer 1 also made similar recommendations. With that in mind, we have made substantial improvements to several of the figures, as follows. We have increased overall figure size and/or font sizes throughout, to ensure that all captions are readily legible. We have simplified Figure 1 to enlarge the pie chart, while omitting detail that can be found in Table 1. In Figure 6, we have increased line thickness to make trends more vivid, and we have labelled each month on the x axis. In Figure 10, we have expanded the abbreviation “Nat Gas” to read “Natural Gas”. In Figure 12, we have increased the size of the lower panel to make the graphs easier to read. We have increased resolution of all graphic files to comply with the journal’s specifications.

We believe that the revised manuscript has been significantly improved. We hope that you agree, and we thank you for pushing us to make our manuscript stronger and clearer.

In addition to the improvements listed above, we have conducted a thorough copy-editing review throughout, to ensure consistency of style and compliance with the specifications of the journal.

Round 2

Reviewer 1 Report

The Authors have referred to all submitted comments, introducing appropriate corrections in the text. The raised doubts have been dispelled, and the text is suitable for publication in its current form.